# Assessing Water Resources Vulnerability by Using a Rough Set Cloud Model: A Case Study of the Huai River Basin, China

**DOI:** 10.3390/e21010014

**Published:** 2018-12-24

**Authors:** Yan Chen, Yazhong Feng, Fan Zhang, Lei Wang

**Affiliations:** 1College of Economics and Management, Nanjing Forestry University, Nanjing 210037, China; 2School of Renewable Natural Resources, Louisiana State University, Baton Rouge, LA 70803, USA; 3Teachers and Teaching Development Center, Nanjing University of Information Science and Technology, Nanjing 210044, China

**Keywords:** Huai River basin water system, water resources vulnerability evaluation, key vulnerable factor identification, rough set-cloud model

## Abstract

Assessing water resources vulnerability is the foundation of local water resources management. However, as one of the major water systems in China, there is no existing evaluation index system that can effectively assess water resource vulnerability for the Huai River basin. To address this issue, we identified key vulnerability factors, constructed an evaluation index system, and applied such system to evaluate water resources vulnerability for the Huai River basin empirically in this paper. Specifically, our evaluation index system consists of 18 indexes selected from three different aspects: water shortage, water pollution, and water-related natural disaster. Then, the improved blind deletion rough set method was used to reduce the size of the evaluation index while keep the evaluation power. In addition, the improved conditional information entropy rough set method was employed to calculate the weights of evaluation indexes. Based on the reduced index system and calculated weights, a rough set cloud model was applied to carry out the vulnerability evaluation. The empirical results show that the Huai River basin water resources were under severe vulnerability conditions for most of the time between 2000 and 2016, and the Most Stringent Water Resources Management System (MS-WRMS) established in 2012 did not work effectively as expected.

## 1. Introduction

Water is one of the most essential natural resources for a country because it is vital for all living creatures to survive. With a per capita water resource of merely a quarter of the global average, China is facing a serious water shortage problem, not to mention the extreme uneven temporal and spatial water distribution within the country. In terms of temporal distribution, most precipitation in China occurs in the summer, while there is usually a lack of precipitation in winter over most of the country. On the other hand, in terms of spatial distribution, the surface runoffs of many rivers in North China have dried up, but many rivers in the South are facing serious flooding problem. Therefore, it is no exaggeration to say that the water systems of many river basins in China are vulnerable, though the reasons for those vulnerabilities vary. Vulnerability of basin water resources is an important frontier topic in the field of water resources research [1]. This type of study can not only measure the water resources vulnerability for a certain river basin quantitatively, but also provide an objective angle to reflect the local water safety situation. Specifically, there are many different reasons for the water resources of a river basin to be consider vulnerable, such as floods, water shortages, water pollution, etc. Different river basins have their own watershed and regional characteristics, which make the main causes for their water resources vulnerability be different from each other. For example, the vulnerability issues of some rivers in Northern China, i.e., the Songhua River, Liao River, and Hai River, are mostly caused by water shortages and water pollution. Meanwhile, for some rivers in the South, such as the Yangtze River, the primarily reason for water resources vulnerability is flooding. Therefore, given the important role that water resources vulnerability assessment plays in water resources management, it will be very meaningful to assess the water resources vulnerability and identify the key vulnerability factors for various major river basins in China.

The research on water resources vulnerability initially emerged from the field of groundwater resources studies as Albinet put forward the concept of vulnerability of groundwater resources in early 1970s [2]. Since then, many scholars and research institutions have begun to study the concept and methodology of groundwater vulnerability evaluation. Since 1990s, the vulnerability of surface water and water systems has gradually become the focus of the literature. For example, Brouwer analyzed the supply and demand balance of regional water resources, as well as the degree of vulnerability of water resources by using threshold of influencing factors [3]. Mirauda used an integrity model as a decision support system to evaluate the vulnerability of surface water resources and used the model to conduct an empirical study of the Bacchiglione basin in Northern Italy [4]. In 1996, the Intergovernmental Panel on Climate Change (IPCC) linked the vulnerability of water resources with climate change and considered that vulnerability was an extension of the damage or adverse effects of climate change on the water resources system [5]. Since then, many scholars began to study the vulnerability of water resources based on the concept proposed by the IPCC. For example, Zhou explored the water resources vulnerability of Chinese cities under climate change conditions [6]. Based on the coupling of natural and social systems, Farley has explored the impact of climate change on water vulnerability in the Oregon mountain area of the United States [7]. Xia’s team studied the impact of climate change on the vulnerability of water resources in the monsoon region of Eastern China and the vulnerability of water resources in the Hai River Basin and its adaptive regulation system [8,9]. Li and Wang studied the vulnerability assessment of water resources system and the adaptive water management system with climate change in Poyang Lake basin [10]. There are some existing studies that have addressed the topic of water resources and vulnerability assessment in the Huai River basin. For instance, Baubion and Marsily used remote sensing technology and integrated hydrological modelling to evaluate water resources in the Huai River basin [11]. Chen and Xia used the Risk-Exposure-Sensitivity-Adaptability model and diversity indexes to assess the Cross-Scale water resource vulnerability and spatial heterogeneity in the Huai River basin [12].

In general, there are two primary quantitative methods to evaluate water system vulnerability: function method and index method. A representative model for function method is the model presented by Xia’s team [13]. The function model mainly focuses on characteristics of the physical mechanism of water resources vulnerability. The original function model considers the vulnerability of water resources system as the ratio of sensitivity to compression resistance [14]. Since then, the Xia’s team have improved the function model according to the definition of the vulnerability of water resources by the IPCC fourth assessment report, adding exposure degree, disaster risk, sensitivity, and compression resistance into the vulnerability assessment model [7]. In addition, other scholars have constructed the vulnerability assessment function models from the perspectives of water supply-demand balance and multi-scale [15,16]. Unlike the function model, the index method constructs an evaluation index system mainly based on the connotation of water resources vulnerability, the influencing factors of water resources vulnerability, DRASTIC model, and Drive, Pressure, State, Impact and Reaction (DPSIR) concept model [17,18,19,20]. Then, the weights of the evaluation indexes are determined mainly by the means of an Analytic Hierarchy Process (AHP), principal component analysis, or entropy weight [21,22,23]. In addition to the aforementioned approaches, fractal theory, fuzzy matter-element evaluation model, the projection pursuit model, and set pair analysis are also used to evaluate the water resources vulnerability [24,25,26,27]. In addition, some database management system and artificial intelligence (AI) predictive models also provide the basis for water resources management. Some scholars have integrated AI method into the flood management system for reservoirs and believe this would be the future trend of flood management [28,29]. In particular, those applications of AI methods include artificial neural network (ANN), particle swarm optimization, and support vector machine (PSO–SVM) model based on ensemble empirical mode decomposition (EEMD) and meta-heuristic technology. Those method were used to forecast rainfall and runoff, which provide some evidences of applying those techniques in water resources management [30,31,32,33].

There are several trending developments in the water resources vulnerability research literature. First, the research objects have been extending from groundwater to surface water and then the whole water resources system. Second, the index system to assess water resources vulnerability has gradually developed from solely water quality research to a combination of water quality and volume of water resources. Third, more scholars are conducting researches on the vulnerability of water resources under the influence of both human activities and climate change. As an important part of the whole water resource system, the river basin is also the main administrative unit of water resources management in China. Therefore, the evaluation of basin water resources vulnerability plays an important role in the decision-making process of local water resources management and planning. Meanwhile, the causes and forms of water resources vulnerability are different over each river basins in China due to the uneven distribution of precipitation and population, as well as the dissimilar industrial layouts over different river basins. In general, the vulnerability of water resources can be separated into several vulnerability factors, such as water shortage, water pollution, and water-related natural disaster, i.e., flood or drought. The evaluation and identification of these key vulnerability factors of water resources in each river basin can help trace the local water resources vulnerability situation and provide guidance for adaptive water management [34]. However, the key vulnerability factor identification for the Huai River basin has not been done to our knowledge. Therefore, the key vulnerability factor identification of basin water resources has become an urgent problem to be solved in the Huai River basin management. The main contributions of this work over existing literature are listed as follows: firstly, this study constructs an evaluation index system of water resources vulnerability from three aspects: water shortage vulnerability (WSVI), water pollution vulnerability (WPVI), and water-related natural disaster vulnerability (WDVI), which are crucial to key vulnerability identification. Secondly, it is a new attempt to apply the method of rough set-cloud model to the assessment of basin water resources vulnerability. Thirdly, in the case study, this study identifies the key vulnerability of water resources in the Huai River Basin, which has not been done by anyone else in the past.

The main purposes of this paper are to evaluate the water resource vulnerability and identify the key vulnerability factors for the Huai River basin water resources quantitatively. The context of paper is organized as follows: in the methodology part, the evaluation index system of water resources vulnerability is constructed to identify the key vulnerability factors of river basin. Specifically, we established a water resources vulnerability evaluation approach combining rough set methods and cloud model. An improved blind deletion rough set method was used to reduce the dimension of evaluation index, and improved conditional information entropy rough set was used to determine the weights of evaluation indices. This approach can be universally applied to any river basin. Afterward, in the case study part, an empirical study was carried out to calculate the vulnerability degree of the Huai River basin water resources using our constructed evaluation index system between 2000 and 2016. Specifically, the identification of water resources vulnerability factors may help clarify the key factors threatening the water resources safety of the Huai River basin. Furthermore, the selection of study period also enables us to compare the water resources vulnerability before and after 2012, at which year the Most Strict Water Resources Management System (MS-WRMS) proposed by the Ministry of Water Resources of China in 2012 was taken into effect. This comparison will reveal how this policy changed the local water resources vulnerability in the Huai River basin. In the last section, discussions are presented, and conclusions are drawn.

## 2. Methodology

In this paper, we firstly propose an evaluation index system for water resources vulnerability evaluation. This index system contains factors covering three different aspects: water shortage, water pollution, and water-related natural disasters. Then we carried out a dimension reduction on the full evaluation index system by using the improved blind deletion rough set method. Afterward, we used the improved conditional information entropy rough set method to determine the weights of indexes in such evaluation index system. Finally, based on the cloud model, we have evaluated the water resources vulnerability of the Huai River basin, measured by calculated water resources vulnerability grading. Since we used rough set type approaches to conduct both weight determination and dimension reduction, and the final evaluation output were derived based on a cloud model, we name the general approach we use for this study as a rough set cloud model.

### 2.1. Construction the Evaluation Index System of River Basin Water Resources Vulnerability

As we have mentioned in the Introduction section, there are two types of mainstream quantitative methods to conduct water vulnerability evaluations: the function approach and the evaluation index method. A valid function approach should have a clear operative mechanism. Thus, it is easy to apply over different regions. In addition, the uniform output format of such approach would allow one to directly compare evaluations results from different regions. However, it is very difficult that one can establish such an ideal function to do the water resource vulnerability evaluation. In addition, in this study we focus on evaluating the water resource vulnerability of the Huai River basin, the comparison over regions, thereby, is not within our scope here. Therefore, we choose the evaluation index approach over the function approach as our basis methodology for this study. First, according to the causes and forms of water resources vulnerability, we separate the water resources vulnerability into three major categories: water shortage vulnerability (WSVI), water pollution vulnerability (WPVI), and water-related natural disaster vulnerability (WDVI). Although the factors may only fall into one single category, they usually work interactively when evaluating water resources vulnerability. Furthermore, each major category contains a second layer of three indexes of vulnerability of water resources: natural vulnerability, man-made vulnerability, and the vulnerability of carrying capacity. Moreover, to assess the water resources vulnerability into greater details, each second layer index is made up with two evaluation variables. In total, we have selected 18 evaluation variables in our evaluation index system. The input data sources of the evaluation index system include the following: “The Water Resources Bulletin of the Basin”, “The China Environmental Statistics Yearbook”, and “The Chinese Statistical Yearbook”. The detail of such evaluation index system is shown in Table 1. 

### 2.2. Improved Blind Deletion Rough Set Method

Given the large number of attributes in a comprehensive evaluation index system, we performed a dimension reduction on the full index system by using improve blind deletion rough set method. Rough set is a mathematical tool dealing with uncertainty, which was initially proposed by Pawlak [35]. A main purpose of this step is to derive a reduced knowledge subset to support the decision-making process, while keeping the same or similar accuracy of original data and eventually increases the efficiency of doing evaluation. Compared to common dimension reduction approaches such as principal component analysis (PCA) and analytic hierarchy process (AHP), improved blind deletion rough set approach can achieve a perfect balance between explanation power and subjective selection. Specifically, the PCA approach is well known as able to greatly reduce the dimension of data while keep its evaluation power. However, PCA uses components to replace the raw attributes in evaluation, so that there is no way to interpret the effects of raw attributes explicitly. On the other hand, the quality of an AHP approach is entirely relies on experts’ judgement, which is somewhat too subjective and may not be reliable all the time. The improved blind deletion rough set method is not only able to reduce the data dimension effectively and objectively, but also allow one to retain some necessary attributes in the system subjectively. Since attributes remain as their original format, the reduced index system produced by improve blind deletion rough set approach can be interpreted in the same way as the full index system. Furthermore, the reduced index system can also be combined with any weight determination methods and evaluation model conveniently.

In this paper, the original data obtained from the study area is discretized to form a decision table according to the grading standard of each index. Then, we use the improved blind deletion rough set method to reduce the dimension of the conditional attributes (evaluation index) in the decision table. The main principle of this method is to calculate the importance index of each subset in the original decision attribute, which will enable us to delete some information while meeting our subjective requirements on evaluation accuracy [36]. Specifically, the usage of this method in our paper can be outlined in following five steps: 

*Step 1*: Based on the weight calculation results from entropy approach and similar results in existing literature, four indexes are selected from primary indexes of WSVI, WPVI, and WDVI, which constitute the initial index attribute set *B*.

*Step 2*: Verify whether this equation $pos_{B}\left(D\right)=pos_{C}\left(D\right)$ is valid or not. If yes, we will continue to add the most important attributes of the set {C−B} to the set *B* by expert judgment, until such equality relationship is valid. For this step, we have consulted a total of 10 experts for their judgement. The details of the scoring process and scores of each attribute are attached in Appendix A.

*Step 3*: Verification of the necessity of all attributes $\beta_{i}\;$in set *B*. First, we calculate the positive region $pos_{\left(B-\beta_{i}\right)}\left(D\right)$ of $\left\{B-\beta_{i}\right\}$ set and the positive region $pos_{B}\left(D\right)$ of set *B*. If this equation $pos_{\left(B-\beta_{i}\right)}\left(D\right)=pos_{B}\left(D\right)$ is not established, then $\beta_{i}\;$is necessary. Afterward, we decide whether the attributes are necessary through the calculation of positive region, so that we can get the necessary condition set *N*.

*Step 4*: We calculate the positive region $pos_{N}\left(D\right)$ of set *N* and the positive region $pos_{B}\left(D\right)$ of *B* set. If this equation $pos_{N}\left(D\right)=pos_{B}\left(D\right)$ is not established, We will continue to add any conditional attributes to the collection set *N* in {*B-N*}, until the equation$\;pos_{N}\left(D\right)=pos_{B}\left(D\right)$ is valid. In this way, the set *N* has the same classification ability as the set *B*.

*Step 5*: Following the last step, we also add some conditional attributes in the set *N* in need of the research contents. Therefore, the set *N* is a reduction of the original decision table *C*.

### 2.3. Weight Determination Approach for the Reduced Evaluation Index System

To determine the weight of evaluation index, we have tried three methods, i.e., the traditional rough set, the conditional information entropy rough set, and an improved conditional information entropy rough set in our pilot study [37,38]. All these three methods are based on rough set theory. Compared to common weight determination approaches such as AHP, using rough set-based approach to determine the weights can effectively avoid the impacts of biased subjective judgements.

The technical details of all these three methods are presented in the following three subsections:

(1) Weight calculation method based on traditional rough set

**Definition** **1.**
*In the decision table S=(U,C,D,V,f), the degree of dependence of decision attribute D on conditional attribute set C is:*
(1)$$\gamma_{B}\left(D\right)=\left|POS_{B}\left(D\right)\right|/\left|U\right|$$


**Definition** **2.**
*In the decision table S=(U,C,D,V,f), c∈C,The important degree of conditional attribute (index) c is defined as:*
(2)$$Sig\left(c\right)=\gamma_{C}\left(D\right)-\gamma_{C-\left\{c\right\}}\left(D\right)$$


The weight W(c) of the conditional attribute c is defined as: (3)$$W\left(c\right)=\frac{Sig\left(c\right)}{\sum_{c\in C}\;Sig\left(c\right)}$$

(2) Weight calculation method based on conditional information entropy rough set

**Definition** **3.**
*In the decision table S=(U,C,D,V,f), U contains any set of attributes S⊆C∪D, which is a random variable on the algebra defined on the subset of U. The probability distribution of S can be determined by the following formula:*
(4)$$\left[S:p\right]=\left[\begin{array}{cccc} S_{1} & S_{2} & \cdots & S_{4}\\ p\left(S_{1}\right) & p\left(S_{2}\right) & \cdots & p\left(S_{4}\right) \end{array}\right],p\left(S{\;}_{j}\right)=\left|S_{j\;}\left|/\right|U\right|,j=1,2,\ldots ,t.$$


**Definition** **4.**
*In the decision table  S=(U,C,D,V,f), I(D|C) is the conditional information entropy of the decision attribute set $D\left(U/D\right)=\left\{D_{1},D_{2},\ldots ,D_{k}\right\}$ relative to the conditional attribute set $C\left(U/C\right)=\left\{C_{1},C_{2},\ldots ,C_{m}\right\}$. The formula for I(D|C) is shown below:*
(5)$$I\left(D/C\right)=\overset{m}{\underset{i=1}{\sum }}\frac{{\left|C_{i}\right|}^{2}}{{\left|U\right|}^{2}}\overset{k}{\underset{j=1}{\sum }}\frac{\left|D_{j}\cap C_{i}\right|}{\left|C_{i}\right|}\left(1-\frac{\left|D_{j}\cap C_{i}\right|}{\left|C_{i}\right|}\right)$$


**Definition** **5.**
*In the decision table S=(U,C,D,V,f), c∈C, The important degree of conditional attribute c is defined as:*
(6)NewSig(c)=I(D/(C−{c}))−I(D/C)+I(D/{c})


The weight W(c) of the conditional attribute c is defined as:(7)$$W\left(c\right)=\frac{NewSig\left(a\right)\;}{\sum_{a\in C}NewSig\left(a\right)}$$

(3) Weight calculation method based on improved conditional information entropy rough set

**Definition** **6.**
*In the decision table S=(U,C,D,V,f), ∀c∈C, μ(c) is the degree of precedence of conditional attribute (index) c:*
(8)$$\mu \left(c\right)=\{\begin{array}{cc} \underset{a\in \{x|x\in C,Sig\left(x\right)=0\}}{\max\left(NewSig\right)} & Sig\left(c\right)\neq 0\\ 0 & Sig\left(c\right)=0 \end{array}$$
*In Equation (8),*
NewSig
*is calculated based on conditional information entropy.*


**Definition** **7.**
*In the decision table S=(U,C,D,V,f), ∀c∈C, where NewW(c) stands for the weight of conditional attribute (index) c:*
(9)$$NewW\left(c\right)=\frac{NewSig\left(c\right)+\mu \left(c\right)}{\sum_{a\in C}NewSig\left(c\right)+\mu \left(a\right)}$$


Among above three weight determination approaches, we finally choose the improved conditional information entropy rough set approach over the other two. The rationale is presented as follows: First, since we are determining the weight for a reduced index system produced by the improved blind deletion rough set method, traditional rough set approach will automatically give consider some attributes as redundant and thereby assign zero weight to them, which include some of those attributes that were considered redundant by the model by manually retained by us. In another word, the traditional rough set weigh determination approach will delete some attributes that we intend to keep in the system. Therefore, we cannot use the traditional rough set approach.

Second, as we can see from Equations (4)–(7), the conditional information entropy rough set approach will eventually produce the output with the weights of redundant attributes are greater than those considered non-redundant. This certainly contradicts what we are intended to do.

### 2.4. Cloud Model

In this paper, a cloud model is used to calculate the vulnerability grading of river basin water resources. Cloud model is a mathematical model to carry out data transformation of uncertain knowledge both qualitatively and quantitatively. As a combination of fuzzy mathematics and traditional probability theory, the cloud model approach was initially proposed by Li [39]. In existing literature, cloud model approach has been successfully applied in many fields, such as intelligent control, machine learning, data mining, system safety assessment, time series prediction, and so on.

#### 2.4.1. Cloud Model and Normal Cloud

The cloud model has three key concepts: cloud space, cloud, and cloud droplet. In this study, we define the whole evaluation index system as a cloud space, which consists of several clouds. Specifically, the grading criterion for each index is regarded as the cloud of the evaluation index system, which represents the correlation between quantitative and qualitative concepts. Moreover, each cloud is made up of several cloud droplets, which are the actual values of indexes. Each cloud droplet will be mapped to the degree of vulnerability in the qualitative concept *C*, where cloud droplets have no order. There will be a random variable with stable tendencies *u(x)* ∈ [0, 1], which is known as the relative degree of certainty of *x* to *C*. In this way, the generation process of each cloud droplet is mapping the uncertainty relationship between quantitative and qualitative concepts [40].

Normal cloud is the basic cloud model that has been widely applied across fields. The main reason for the popularity of normal cloud is that normal and semi-normal distributions are widely used to describe expected probability density curves of natural and social events. Random variables of a normal cloud follow the normal distribution approximately. A normal cloud can be described by three parameters: the expected value *Ex*, the entropy *En*, and the hyper entropy *He*. The expected value *Ex* is the mean of cloud droplet values in a defined domain, which represents the most typical sample of data. Entropy *En* measures uncertainty belonging to a certain degree of vulnerability, reflecting the variability level and the range of specific index. The hyper entropy *He* is the entropy of entropy *En*. It is the measurement of the uncertainty of entropy and the dispersion of cloud droplets. In addition, the hyper entropy also represents the cohesiveness of index values in the discoursed domain. For example, a large value of hyper entropy means the value of evaluation index are more likely to spread out from its expected value, and the cloud will be relatively thick. Figure 1 shows a diagram for normal cloud and its parameters.

#### 2.4.2. Forward Direction Generator of Cloud Model

The normal cloud generator is based on randomly generalized data following a normal distribution. The quantitative numerical interval of water resources vulnerability evaluation is defined as *U*, which is called discourse domain. *C* is the quantitative grading of river basin water resource vulnerability corresponding to the *U*. The value of evaluation index of water resources vulnerability in discourse domain *U* is *x*, *x* ∈ *U*, and *x~N*(*E_x_*, *E’_n_*^2^), where *E_x_* is the expected value of *x*, and *E’_n_* is a normal random variable. The relative degree of certainty of *x* relative to *C* is *μ*, which can be obtained by the forward direction generator of given cloud model. Following this method, we can generate cloud droplets based on the normal cloud with certain characteristic parameters (*Ex*, *En*, *He*) to get a quantitative mapping relationship. The specific steps are shown below:

*Step 1*: Based on the collected dataset of water resources vulnerability evaluation index, we calculate the normal cloud characteristics (*Ex*, *En*, *He*).

*Step 2*: Based on the characteristic parameters calculated in last step, we generate a random sample of interested random variable *x* as *x~N*(*E_x_*, *E’_n_*^2^), and *E’_n_*~(*E**_n_*, *He*^2^). 

*Step 3*: Based on the random sample generated in last step, a cloud droplet can be generated by calculating the relative degree of certainty of each evaluation index based on Equation (10).
(10)$$\mu =\exp\left[\frac{-{\left(x-\;E_{x}\right)}^{2}}{2E_{n}^{,2}}\right]$$

*Step 4*: Repeat the above three steps to generate enough cloud droplets. In this paper, a total of 15,000 cloud droplets are generated for the case study. 

#### 2.4.3. Calculating the Characteristic Parameters of a Cloud Model

*Ex* is the expected value of vulnerability evaluation index, which is defined by formula:
(11)$$\;Ex_{ij}=\frac{X_{ij}^{1}+\;X_{ij}^{2}}{2}$$

In the formula, $X_{ij}^{1}$ is the lower boundary value, and $X_{ij}^{2}$ is the upper boundary value. *i* (*i* = 1, 2, …, *n*) is an indicator, and *j* (*j* = 1, 2, …, *m*) is a grade.

The entropy *En* of the cloud model is calculated by Equation (12):(12)$$En_{ij}=\left(X_{ij}^{1}-\;X_{ij}^{2}\right)/2.355$$

Hyper entropy *He* = *k*, where *k* is a constant, which reflects the dispersion degree of the cloud model. We can adjust the value of *k* according to the fuzzy threshold of variables. In this paper, we take a unified value and define *k* = 0.01.

#### 2.4.4. Steps of Vulnerability Evaluation Using Cloud Models

In this paper, we construct a methodology to assess the water resources vulnerability as a combination of the improved blind deleting rough set method, the improved entropy rough set method, and the cloud model. The flowchart of the evaluation process is presented in Figure 2.

*Step 1*: Establish the evaluation index system (WSPD-VI), which is constructed from three aspects: water shortage vulnerability, water pollution vulnerability, and water-related natural disaster vulnerability.

*Step 2:* We use the improved blind deletion rough set method to reduce the dimension of the evaluation index and remove the redundant attributes.

*Step 3*: The improved conditional information entropy rough set method is used to determine the index weights in the evaluation index system. Through the comparison of the three methods, we calculate the weight matrix *W* according to the formulas in Definitions 6 and 7.

*Step 4*: Each evaluation index has five grades: extreme vulnerability, severe vulnerability, moderate vulnerability, mild vulnerability, and no vulnerability. Then, according to the international, domestic, and industry standards of each evaluation index, we determine six thresholds.

*Step 5*: Based on the grading thresholds determined in last step, we use Formulas (11) and (12) to calculate the characteristic parameters of the cloud model. MATLAB was used to program the forward generator of cloud model to determine the degree of each evaluation index and form the matrix *Z* of the cloud model’s certainty degree.

*Step 6*: We use the weight matrix *W* and the degree matrix *Z* to carry out the fuzzy transformation to conduct a comprehensive vulnerability assessment of certain river basin water resources system.

## 3. Case Study

### 3.1. Research Area

The Huai River basin is located in Eastern China, about the midway between the Yangtze River basin and the Yellow River basin. Geographically, the Huai River basin ranges from 111°55′ E to 121°25′ E, and 30°55′ N to 36°36′ N. The Huai River sourced from the Tongbai Mountain on the junction of Tongbai of Henan Province and Suizhou of Hubei Province. The Huai River basin includes 40 cities and 163 counties in the five provinces of Hubei, Henan, Anhui, Jiangsu, and Shandong, with a total area of approximately 27 × 10^4^ km^2^. About 2/3 of the total area inside the Huai River Basin is a plain area, while some mountainous and hilly areas accounting for 1/3 of the total river basin area are scattered in the western, southwest, and northeastern parts of the river basin. The total length of the Huai River is about 1252 km, and the total elevation difference between highest and lowest point is about 200 m. The Huai River basin can be divided into three parts: the upper reaches located between Tongbai Mountain and Hong River Estuary; the middle reaches located between the Hong River Estuary and the Hongze Lake, and the lower reaches ranging from Hongze Lake to Sanjiangyin where the Huai River meets the Yangtze River.

Located in the north-south climate transition zone, the Huai River is widely regarded as one of the natural dividing lines between North and South in China. In terms of temperature, the northern part of Huai River basin belongs to the warm temperate climate zone, where the temperature varies greatly over the four seasons of a year. The climate in this area is generally humid and hot in summer while dry and cold in winter. The southern part of Huai River basin belongs to the humid subtropical climate zone, where the climate is mild, with obvious sign of climate transition between North and South [41]. The geographical mapping of the Huai River basin is shown in Figure 3.

With a population density of 690 people/km^2^, which is 4.91 times the national average, the population of the Huai River basin area accounts for 13.5% of the total population of China. The per capita water resource in the Huai River basin is only 373.8 m³, which is only 1/5 per of the national average level of China. At present, the Huai River basin is facing many water problems such as flooding, water shortages, and water pollution. The water resources vulnerability issue is restricting the whole area from furthering ecological and economic development. However, this topic of water resources vulnerability evaluation and key vulnerability factor identification in this region has not been studied in the past. Therefore, there is an immediate need to carry out such a study for the Huai River basin. 

### 3.2. Evaluation Grade and Trend Analysis of Water Resources Vulnerability in Huai River Basin

We use the steps of the model method in Figure 2 to calculate the vulnerability assessment grade of the Huai River Basin.

#### 3.2.1. Attribute Reduction of Water Resources Vulnerability Evaluation Index

(1) Data Sources

We collected evaluation index data for the Huai River basin water resources based on the research period between 2000 and 2016. The data sources included but were not limited to the following: “The Water Resources Bulletin of the Huai River Basin” (2000 to 2016), “The Governance of Huai River” (2000 to 2016), “The China Environmental Statistics Yearbook” (2000 to 2016), and “The Chinese Statistical Yearbook” (2000 to 2016). 

We choose the study period from 2000 to 2016 mainly for two reasons. The first one is the availability of index data, as the availability of many indicators before 2000 is limited. In addition, the Yearbooks of 2017 and 2018 have not been published to date, so the latest data we could obtain is for the year of 2016. The second one is the fact that Chinese Ministry of Water Resources began to implement the MS-WRMS in 2012. This policy is supposed to generate a significant effect in the water resources vulnerability conditions in the Huai River basin, and thus should be incorporated in our research period to enable a test for its efficiency. 

(2) Decision Table

The reduction of evaluation index system by the rough set method needs a decision table *C*, which can be divided into two categories. The first category is the condition attribute, which consists of categorical variables that were discretized from the original data of each index by applying predetermined index thresholds. The other is the decision attribute. We use entropy weight method to determine the weight of the evaluation index. Then we use the cloud model to calculate the water resources vulnerability grades of the Huai River basin between 2000 and 2016 to form the decision table.

(3) Dimension Reduction Process of the Evaluation Index

In this paper, we use the improved blind deletion rough set method to reduce dimension of the evaluation index. According to the steps in Section 2.2, the implementation process is presented as follows:Set up the initial index set *B*. We first select 12 important evaluation indicators from the original index set, that is, $B=\left\{A_{1},A_{2},A_{3},A_{4},B_{1},B_{3},B_{5},B_{6},C_{1},C_{2},C_{3},C_{6}\right\}$Whether the validation equation $POS_{B}\left(D\right)=POS_{C}\left(D\right)$ is set up. We verify that this equation $pos_{B}\left(D\right)=pos_{C}\left(D\right)=\left\{\left\{x_{1}\right\}\left\{x_{2}\right\}\left\{x_{3}\right\},\ldots \left\{x_{33}\right\}\right\}$ is set up. It shows that the initial index set *B* has the same classification ability as the original conditional attribute set *C* and does not need to add additional index.Verify the necessity of each evaluation index in index set *B*. The verification process of each evaluation index in set *B* is as follows:

*Step 1*: Remove the index$\;A_{1}$. The equation $pos_{B-A_{1}}\left(D\right)=pos_{B}\left(D\right)$ is established, indicating that index$\;A_{1}\;$can be removed.

*Step 2*: Remove the index$\;A_{2}$. We calculate $pos_{B-A_{2}}\left(D\right)\neq pos_{B}\left(D\right)$, so indicator $A_{2}\;$is necessary.

*Step 3*: Remove the index$\;A_{3}$. The equation$\;pos_{B-A_{3}}\left(D\right)=pos_{B}\left(D\right)$ is established, indicating that index$\;A_{3}\;$can be removed.

Repeat the above three steps until the data converges. Through the above steps, we conclude that the index set $N=\left\{A_{2},A_{4},B_{3}\right\}$ satisfies condition$\;pos_{B-A_{i}}\left(D\right)\neq pos_{B}\left(D\right)\left(i=1,2\ldots 12\right)$. Then we calculate the $pos_{N}\left(D\right)=19\neq pos_{B}\left(D\right)$, indicating that the remaining indexes in the set *B* can’t be deleted at the same time.
According to the order of importance of the index, we select the important indexes from the set {*B*-*N*} collection to add to the collection set *B*:

*Step 1*: We add $A_{1}\;$to set *N* and calculate the $pos_{(N+}{}_{B_{2})}\left(D\right)\neq pos_{B}\left(D\right),\;\left|pos_{(N+}{}_{B_{2})}\left(D\right)\right|=27$. We thus found that the existing set *N* does not have the same classification ability as the set *B*, so we continue to add other indicators. At the same time, $N=\left\{A_{1},A_{2},A_{4},B_{3}\right\}$.

*Step 2*: We add $B_{6}$ to set *N* and calculate the$\;pos_{(N+}{}_{B_{6})}\left(D\right)\neq pos_{B}\left(D\right),\;\left|pos_{(N+}{}_{B_{6})}\left(D\right)\right|=29$. We found that the existing set *N* does not have the same classification ability as the set *B*, so we continue to add other indicators. At the same time, $N=\left\{A_{1},A_{2},A_{4},B_{3},B_{6}\right\}$.

*Step 3*: We add $C_{1}$ to set *N* and calculate the$\;pos_{(N+}{}_{C_{1})}\left(D\right)\neq pos_{B}\left(D\right),\;\left|pos_{(N+}{}_{C_{1})}\left(D\right)\right|=29$. We found that the existing set *N* does not have the same classification ability as the set *B*, so we continue to add other indicators. At the same time, $N=\{A_{1},A_{2},A_{4},B_{3},B_{6},C_{1}\}.$

*Step 4*: We add $C_{2}$ to set *N* and calculate the $pos_{(N+}{}_{C_{2})}\left(D\right)\neq pos_{B}\left(D\right),\;\left|pos_{(N+}{}_{C_{2})}\left(D\right)\right|=31$. Since we found that the existing set *N* does not have the same classification ability as the set *B*, we continue to add other indicators. At the same time, $N=\left\{A_{1},A_{2},A_{4},B_{3},B_{6},C_{1},C_{2}\right\}$

*Step 5*: We add$\;C_{3}$ to set *N* and calculate the$\;pos_{(N+}{}_{C_{3})}\left(D\right)\neq pos_{B}\left(D\right),\;\left|pos_{(N+}{}_{C_{3})}\left(D\right)\right|=31$. We found that the existing set *N* does not have the same classification ability as the set *B*, thus, we continue to add other indicators. At the same time, $N=\left\{A_{1},A_{2},A_{4},B_{3},B_{6},C_{1},C_{2},C_{3}\right\}$

*Step 6*: We add$\;B_{1}$ to set *N* and calculate the $pos_{(N+}{}_{B_{1})}\left(D\right)=pos_{B}\left(D\right),\;\left|pos_{(N+}{}_{B_{1})}\left(D\right)\right|=33=pos_{B}\left(D\right)$. We found that the existing set *N* have the same classification ability as the set *B*., so we stop adding other indicators. Finally,$\;N=\left\{A_{1},A_{2},A_{4},B_{1},B_{3},B_{6},C_{1},C_{2},C_{3}\right\}$.
Through the above steps, we can get a better reduction index set $N=\left\{A_{1},A_{2},A_{4},B_{1},B_{3},B_{6},C_{1},C_{2},C_{3}\right\}$, which has the same evaluation ability as the conditional attribute *C*. At the same time, the index is more balanced in the process of adding and can be used as the basis of cloud model calculation. We use the improved blind deletion rough set reduction method to get the evaluation index system as shown in Table 2.

In the process of using rough set method to reduce the dimension of index, we have retained several indicators subjectively. Although from the perspective of data mining these indicators are redundant attribute indicators, however, these indicators are very important for the evaluation of basin water resources vulnerability in practice. Therefore, the improved blind deleting method is a method which can fully take advantage of both actual situation and data mining.

#### 3.2.2. Calculation of the Weights of Evaluation Indexes

We use the improved condition information entropy rough set method to determine the weights of the evaluation index system. Here we illustrate how we make this decision on methodology to calculate the weights of evaluation indexes. 

The first approach we have tried to determine the index weight is to use the importance degree attribute of the traditional rough set [42]. By applying this approach, we find that there are three redundant attribute indexes, which are *A*_1_, *C*_1_ and *C*_3_. In fact, the weights of these three indexes cannot be calculated based on traditional rough set. However, as these three indexes are required to be in the model for the evaluation of basin water resources vulnerability, we conclude that this method is not suitable for this study.

The second approach we tried is the conditional information entropy rough set method [43]. The results indicate that the weights of redundant attributes are higher than some non-redundant attributes, which is inconsistent with the theory of rough sets thus we cannot use this approach here.

Finally, we use the improved conditional information entropy rough set algorithm priority to determine the weight of evaluation indexes, as illustrated in Definitions 1 and 2. The final determined weights of the evaluation index system are shown in Table 2.

#### 3.2.3. Calculation of the Characteristic Values and Cloud Model

To determine the grading standard of nine evaluation indicators, we refer to the current international, domestic, and water conservancy standards. The specific technical detail of the evaluation index is shown in Table 3. Specifically, for the indicator *A*_4_, it is worth noting that there is a rule generally accepted internationally that the proportion of water resources that have been developed or utilized should not exceed 40% of its total water resources, and a river with over 55% of such ratio is regarded as extremely vulnerability. 

According to the grading range of each evaluation index in Table 3 and Equations (4) and (5) for calculating the characteristic values of the cloud model, we can calculate the coefficient of each index corresponding to each evaluation grade. Due to space limitation, we do not list the details of calculated characteristic values. The cloud model of the nine evaluation indexes are presented in Figure 4. In Figure 4, I, II, III, IV, V stands for “extreme vulnerability”, “severe vulnerability”, “moderate vulnerability”, “mild vulnerability”, and “not vulnerable”, respectively. 

#### 3.2.4. Degree of Certainty and Trend Analysis of Basin Water Resources Vulnerability 

Our MATLAB program has three main inputs: calculated degrees of certainty, weights of indexes calculated earlier, and the raw data of each index between 2000 and 2016. Then we run the cloud model program for 15,000 rounds and take the means as the results. Finally, according to the principle that the grading is determined by the maximum value of degree of certainty, we can determine the degree of water resources vulnerability in the river basin. Similarly, using the above methods, we calculated the degree of certainty and the grades of water shortage vulnerability, water pollution vulnerability and water-related natural disaster vulnerability. The calculation results are shown in Table 4.

To illustrate the trend of the water resources vulnerability in the Huai River Basin from 2000 to 2016, we draw a diagram as shown in Figure 5. As the trend line shows in Table 4, the degree of water resources vulnerability in the Huai River basin has fluctuated around level 2 during the research period. It shows that the water resources vulnerability in the Huai River basin was under severe vulnerability level for most of the time. On the other hand, the results indicate that there remains large space for the water resources management in the Huai River basin to be improved.

### 3.3. Identification of Key Vulnerability in Huai River Basin

The vulnerabilities of water resources in different river basins in China are caused by different reasons. Only by identifying the key vulnerabilities of each river basin and finding the short board of the water management system of such river basin, we can effectively further the planning and management of river basin water resources. In this paper, an evaluation index system has been established to evaluate river basin water resources vulnerability from the aspects of water shortage, water pollution, and water-related natural disaster. The aim of this study is to figure out the important factors that cause the vulnerability of water resources in the Huai river basin to facilitate the adaptive management. Since the annual vulnerability of the Huai River basin fluctuates over time, here we use annual mean values to analyze the concentration trend. To clearly highlight the key vulnerability of water resources in the Huai River basin comprehensively, we calculated the mean annual values of the WSPD-VI, WSVI, WPVI, and WDVI in the Huai River basin between 2000 and 2016. A radar chart visualizing the water resources key vulnerabilities in the Huai River basin is shown in Figure 6.

The radar chart shows that the average grade of water shortage vulnerability in the Huai River basin is 3.47, followed by the average grade of the water-related natural disaster vulnerability of 2.65, thereafter is the average grade of the water pollution vulnerability of 1.82. According to definition of vulnerability degree, lower the vulnerability grade indicates more vulnerable status of the water resources. Obviously, we can find that the key factors causing water resources vulnerability of the Huai River basin is ranked by water pollution, followed by water-related natural disaster, and water shortage is of least concern. In the following sections, the details of these three key vulnerabilities in the Huai River basin are to be elaborated in greater details. 

#### 3.3.1. Analysis of the Vulnerability of Water Shortage in Huai River Basin

As mentioned before, the Huai River basin is the climate transition zone of South and North China. The precipitation in the whole area is very uneven, as the precipitation volume in the South is far greater than the North. Since rainfall is the most important source of water in the river basin, the water resources are not evenly distributed over the whole region. Among the three reaches of the Huai River, the one that has highest level of mean annual precipitation in the upper reach, while the region with the least annual rainfall is the middle reach. There is also significant difference in precipitation levels over the five provinces that the Huai River basin runs through. In addition, the annual precipitation in the Huai River basin also varies greatly chronologically. Historically speaking, the mean annual precipitation over the whole region is approximately 880 mm [44]. However, the actual mean annual precipitation ranges from the lowest record of 300 mm to the highest record year with 1300 mm. In the research period between 2000 and 2016, the water shortage vulnerability of the Huai River basin is seriously imbalanced as shown in Figure 7.

We also analyzed the monthly precipitation variation in the Huai River basin. The monthly precipitation in the Huai River basin also varies a lot. Precipitation is mainly concentrated between June and September of each year, usually accounting for about 50% to 80% of the total annual precipitation. Thus, this period is usually the flood season for the Huai River. On the other hand, winter and spring usually have less rainfall than other seasons. In addition, the water runoff volume of Huai River also shows strong seasonal pattern as the maximum monthly runoff is 30 times of the minimum. Such a huge difference will bring considerable challenges for local water resources management.

To clearly illustrate the trend of vulnerability of water resources shortage in the Huai River basin, we draw the trend diagram includes the trends of water yield per km^2^ (*A*_1_), absolute value of variation coefficient of annual precipitation (*A*_2_) and the proportion of groundwater resources being utilized (*A*_4_) from 2000 to 2016, as shown in Figure 8.

As shown in Figure 8, all three of these indicators were fluctuating in the 17-year research period. In detail, *A*_1_ and *A*_2_ show relatively consistent patterns, while *A*_4_ seems to be negatively correlated with these two indicators. This fact reflects that when the water yield of the Huai River basin is large, there is no need to exploit the groundwater resources too much. However, during the period between 2008 to 2015 when the water yield is relatively low, more than 40% of groundwater resources in the Huai River basin was exploited, which is higher than the international warning level.

In addition, we found that the precipitation and total water resources in the Huai River basin were decreasing during the period between 2000 to 2016. Specifically, the change rate of precipitation is −51.43 billion m^3^/5a, and the change rate of total water resources is −20.45 billion m^3^/5a. Through the analysis of the river basin water shortage vulnerability, it can be found that although the annual average precipitation is relatively high, more than half of our whole study period was characterized as moderate to severe water shortage. Obviously, planning and management for the water resources utilization in the Huai River basin are challenging jobs. This is mainly due to three reasons: the large variation of precipitation level year to year, the uneven distributed precipitation within any calendar year, the uneven spatial distribution of precipitation over the whole river basin. 

#### 3.3.2. Analysis of the Vulnerability of Water Pollution in the Huai River Basin

The Huai River’s watershed drains five provinces, 40 cities, with a total river drainage basin area of approximately 2.747 × 10^4^ km^2^, the whole region is primarily a flood plain. With a total population of more than 180 million, this region is one of the most densely populated areas in China, as well as one of the three major crop growing regions in China [45]. Since 1980s, the Huai River has undergone a series water pollution problem mainly due to the boom of small manufacturing businesses, rapid urbanization and population growth, lack of environmental protection, and poor water management. In fact, large amount of untreated industrial sewage had been discharged directly into the Huai River and its tributaries. In addition, the growing usage of fertilizers and pesticides in agriculture also serves as a major source of water pollution in the Huai River basin. Since water pollution had become a serious problem in this region, the surface water is barely able to be utilized directly. Thus, people were forced to exploit groundwater. In fact, the shallow groundwater levels in some cities were significantly decreased due to the excessive exploitation of groundwater. For the same reason, salt water intrusion along the coastline, aggravating the local water resources shortage and ecosystem imbalance are also emerging in this region.

According to the data from the Ministry of Environmental Protection of China, the Huai River is the most polluted water system among the seven major water systems in China. The main pollutants include ammonia nitrogen, organic compound, potassium permanganate, and phosphorus. We draw a trade graph of the vulnerability of water pollution-related vulnerability grading in the Huai River basin over the period between 2000 and 2016 in Figure 9. It as what we can see from the chart, the water pollution vulnerability in the Huai River basin was under serious or extreme vulnerability grade for most of the research period. It can also be seen that the Huai River basin was facing a serious water pollution problem, which serves as a key factor contributing to the overall vulnerability of the water resources in the Huai River basin.

The pollution of the Huai River has drawn the national attention and it has become one of the major environmental protection challenges in China. To protect the local living environment from further worsening, the Huai River basin has become the first river basin where a comprehensive management action was taken place in the country. Since 1990s, both Chinese national and local governments have devoted meaningful amount of resources to establish the water quality monitoring system and water pollution prevention and control project. So far, considerable progress has been made on resolving the water pollution problem in the Huai River basin. To reveal the effectiveness of the water pollution control efforts, the trend charts of several water pollution indexes are shown in Figure 10 and Figure 11. Figure 10 presents the trend of the water quality examination pass rate in water function area (*B*_1_) over the research period, which basically shows a two-stage pattern. Specifically, the index *B*_1_ was declining before 2007, and then turned around to be getting better over time. However, until the end of this study, the water quality pass rate in the Huai River basin has never reached 70%. This is still lower than a targeted rate of 80%. Specifically, this target was originated from a proposal by the Ministry of Water Resources of China that the water quality examination pass rate should be greater than 80% by 2020 in the functional areas of the major rivers and lakes in China. 

Figure 11 demonstrate that how the amount of wastewater generation per 10,000-yuan GDP(*B*_6_) changed over the research period from 2000 to 2016, which is obviously a declining trend. This also shows that the Huai River basin has made some progress in water pollution control as the wastewater control measures indeed worked. Although the water quality of the Huai River basin has been improved as the amount of pollutant discharge was significantly reduced, they are way below the targeted water quality level. In fact, the amount of pollutants discharged into the waterbody still exceeds the environmental capacity. On the one hand, the water quality of the Huai River largely relies on natural runoff dilution. However, the peak of pollutant discharge usually appears in the dry season when the natural runoff is low, which makes the water quality of the river even worse for some time periods. Moreover, the dams on the river also cause the pollutants to be accumulated nearby. When the dams discharge during flood season, the accumulated pollutants would also be discharged to lower reaches, which causes a man-made peak of water pollution level temporarily exceeding the environmental capacity. Although such temporary water pollution will not last long, it still causes serious problems that are difficult to recover [46]. 

#### 3.3.3. Analysis of the Water-related Natural Disaster Vulnerability in the Huai River Basin 

The Huai River basin is widely regarded as a region suffering from both flood and drought disasters for some geographic and climate reasons, primarily the uneven precipitation over regions and seasons. As the most rainfall is concentrated between June and September, rainwater can easily gather quickly to form flooding runoffs. On the contrary, as the precipitation level is far lower during winter, it is also easy for this region to be under drought during this season. In general, the frequency of flood is almost once a year, and the average interval between droughts is approximately 1.7 years in the Huai River basin. In addition, it is also common to see the flood and drought to be happened within the same calendar year. For example, severe spring drought and three floods occurred in Anhui Province inside the Huai River basin in 2009.

The trend of water-related natural disaster vulnerability in the Huai River basin during the period of 2000 to 2016 is presented in Figure 12. As we can see from the graph, the disaster defense capability of the Huai River basin has been gradually improving over the past 17 years. After nearly 50 years of unremitting efforts, a large number of water conservancy projects have been built in the Huai River basin to form a relatively comprehensive system for flood control, water storage, and irrigation. Those projects have greatly improved the capability of water-related natural disaster prevention [47].

To further demonstrate the improvement on water-related natural disaster preparation in the Huai River basin, the trend of proportion of area affected by flood and drought (*C*_2_) over the last 17 years is shown in Figure 13. We can see that although the trend of this index is fluctuating over the past 17 years, but there is basically a downward trend. It shows that the region suffering from flood and drought has been reduced gradually during our research period as a result of large amount of water conservancy projects being built. Currently, the Huai River basin has more than 5700 reservoirs with a total storage capacity of nearly 27 billion m^3^. Among them, there are 36 large reservoirs with a total capacity of 18.7 billion m^3^. In addition, the Huai River basin has all kinds of dams with a total length of more than 50,000 km. Moreover, there are a couple of wetlands and lowlands along the Huai River with a total area of about more than 4000 km^2^, which also make up an important part of the flood control project system because those can be utilized as flooding buffer zones.

### 3.4. Discussion of the Main Results

In this section, we discuss the following five indexes: WSPD-VI, WSVI, WPVI, WDVI and the key vulnerability. The overall degree of water resources vulnerability (WSPD-VI) in the Huai River basin has been fluctuating around level two during our research period from 2000 to 2016. This shows that the water resources vulnerability in the Huai River basin was under severe vulnerability level for most of the time. Furthermore, after the MS-WRMS went into effect in 2012, the vulnerability of basin water resources has not been improved significantly. A possible reason for this counterintuitive fact is that it usually takes time for water conservation measures and policies to generate impact to local water system. On the other hand, the results indicate that there remains large room for the water resources management to be improved in the Huai River basin.

The water shortage vulnerability (WSVI) of the Huai River basin is seriously imbalanced over the research period. Through the analysis of the WSVI, it can be found that although the annual average precipitation is relatively high, i.e., more than half of the entire study period was characterized as moderate to severe water shortage. This is mainly due to an uneven distributed precipitation within any calendar years. Historically speaking, the mean annual precipitation over the whole region is approximately 880 mm. However, the actual mean annual precipitation ranges from the lowest record of 300 mm to the highest record of 1300 mm. In another word, the maximum annual precipitation is 4.3 times greater than the minimum recorded level, which indicates huge imbalance. Obviously, planning and balancing the water resources utilization over time in the Huai River basin are challenging jobs.

The water pollution vulnerability (WPVI) in the Huai River basin was under severe or extreme vulnerability grade for most of the research period. Although the water quality of the Huai River basin has been improved as the amount of pollutant discharged was significantly reduced, they are way below the targeted water quality level. In fact, the amount of pollutants discharged into waterbody still exceeds the environmental capacity. According to the Statistics of China’s environmental situation in 2016, 46.7% of the total waterbody in Huai River basin have been categorized into level IV, V, or poor V. These statistics indicate that the water pollution in the Huai River basin remain severe and it needs further treatment. The Huai River basin was facing a severe water pollution problem, which serves as a key factor contributing to the overall vulnerability of the water resources in the Huai River basin.

The trend of water-related natural disaster vulnerability (WDVI) in the Huai River basin has been gradually improved during the period of 2000 to 2016. The main reason is that the large number of water conservancy projects that have been built in the Huai River basin form a relatively comprehensive system for flood control, water repository, and irrigation in the past 50 years. Those projects have greatly improved local capability to overcome water-related natural disasters. However, the Huai River basin is widely regarded as a region suffering from both flood and drought disasters for some geographic and climate reasons, primarily the uneven precipitation over regions and seasons. Watershed Management Committees need to continue to pay attention to the prevention of flood and drought.

The radar chart shows that the average grade of water shortage vulnerability in the Huai River basin is 3.47, followed by the average grade of the water-related natural disaster vulnerability of 2.65, thereafter is the average grade of the water pollution vulnerability of 1.82. According to definition of vulnerability degree, lower the vulnerability grade indicates more vulnerable status of the water resources. Obviously, we can find that the key factors causing water resources vulnerability of the Huai River basin is ranked by water pollution, followed by water-related natural disaster, and water shortage is of least concern.

## 4. Conclusions

In this paper, we established a rough set cloud model to evaluate the river basin water resources vulnerability and identified the key vulnerabilities for a river basin. Then we use the model to carry out an empirical case study on the Huai River basin water resources vulnerabilities. The main conclusions are presented in this section.

Basin water resources systems are a complex system. The causes and forms of the water resources vulnerability vary over different river basins as each one has its own characteristics. At first, based on the causes and forms of water resources vulnerability, we set three primary indexes from three aspects: water shortage vulnerability (WSVI), water pollution vulnerability (WPVI), and water-related natural disaster vulnerability (WDVI). Secondly, based on the factors affecting the vulnerability of water resources, we further divide each primary index into three sub layer indexes: natural vulnerability, man-made vulnerability, and the vulnerability of carrying capacity. Finally, we selected 18 evaluation indicators to form a comprehensive water resources vulnerability evaluation index system. We name this system WSPD-VI, which enable us to identify the key vulnerability of a river basin.

The improved blind deletion rough set method is used to reduce the dimension of evaluation index system. Removal of redundant indicators can make the evaluation index system more concise while keep the evaluation power of the original system. Further, another reason for us to choose the blind deletion rough set method is that it allows us to keep indicators subjectively, which enable us to keep some indicators show low weight in calculation but still necessary. Finally, the cloud model method is used to calculate the vulnerability degree of basin water resources. The cloud model is a method combining the traditional fuzzy mathematics and the probability theory. It can tolerate the uncertainty mapping of each evaluation index to the evaluation grade. By using cloud model to determine the vulnerability grade of river basin water resources, we can consider some randomness imbedded in the environment.

An empirical analysis has been carried out to evaluate the water resources vulnerability in the Huai River basin between 2000 and 2016. The results show that the vulnerability degree of water resources in the Huai River fluctuated around the second level (severe vulnerability) during the study period. In addition, the water resources vulnerability condition did not show significant improvement after the MS-WRMS took into effect in 2012. Specifically, from the key vulnerability identification of the Huai River basin, the average grade of water shortage vulnerability is 3.47, followed by the average grade of the water-related natural disaster vulnerability of 2.65, and thereafter the average grade of the water pollution vulnerability is 1.82. Obviously, in terms of impact, the sequence of key factors affecting the water resources vulnerability of the Huai River basin is water pollution, water-related natural disaster, and water shortage.

The empirical results show that the MS-WRMS that took effect in 2012 by the Ministry of Water Resources of China did not turn around the condition of water resources vulnerability in the Huai River basin as expected. However, the recovery of water resources usually take time. In this study, due to the limit in observations after the implementation of such policy, the ineffectiveness within four years does not really mean that this new policy will not generate any further effect in the future. Due the fact that this problem has to be address when more observations can be collected, it is necessary to assess the water resource vulnerability of the Huai River basin again in the future to gain better understanding on the true effects of this policy.

## Figures and Tables

**Figure 1 entropy-21-00014-f001:**
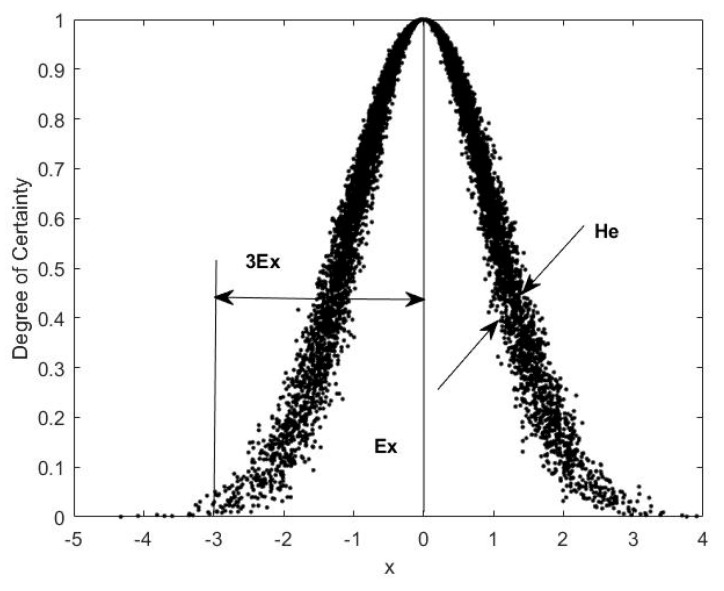
The diagram of a normal cloud and its parametrization.

**Figure 2 entropy-21-00014-f002:**
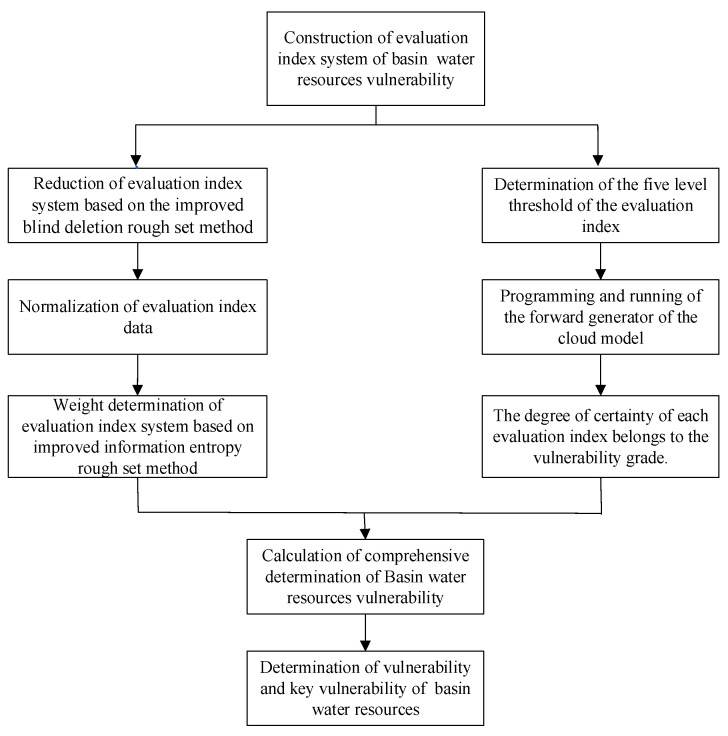
Flow chart of basin water resources vulnerability evaluation model.

**Figure 3 entropy-21-00014-f003:**
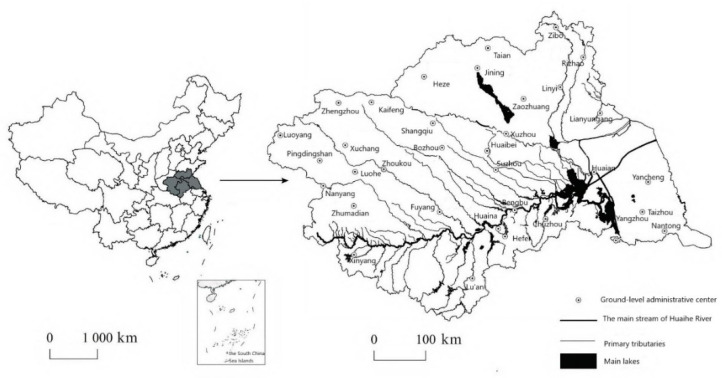
The map of Huai River drainage basin and its water system.

**Figure 4 entropy-21-00014-f004:**
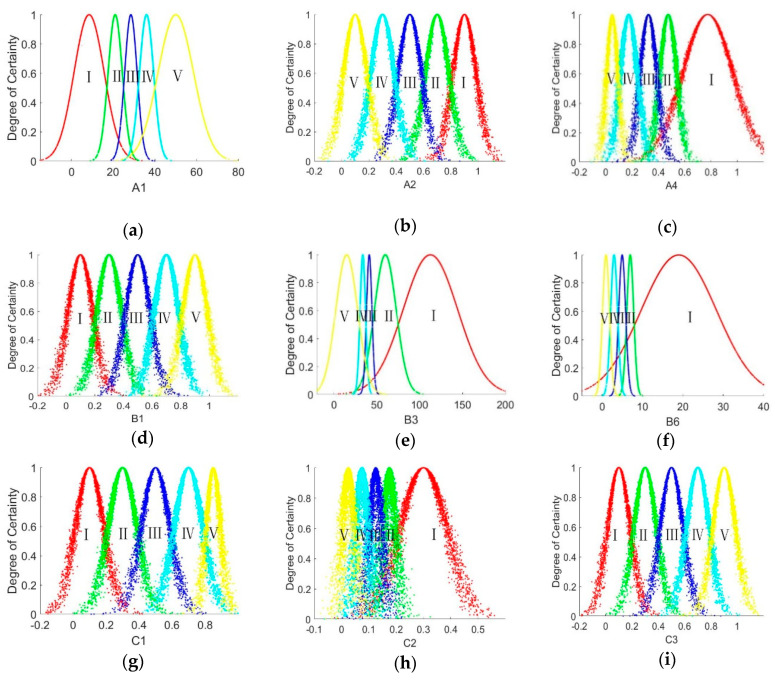
Cloud model of nine evaluation indexes. (**a**) Normal cloud figure of evaluation index *A*_1_; (**b**) Normal cloud figure of evaluation index *A*_2_; (**c**) Normal cloud figure of evaluation index *A*_4_; (**d**) Normal cloud figure of evaluation index *B*_1_; (**e**) Normal cloud figure of evaluation index *B*_3_; (**f**) Normal cloud figure of evaluation index *B*_6_; (**g**) Normal cloud figure of evaluation index *C*_1_; (**h**) Normal cloud figure of evaluation index *C*_2_; (**i**) Normal cloud figure of evaluation index *C*_3_.

**Figure 5 entropy-21-00014-f005:**
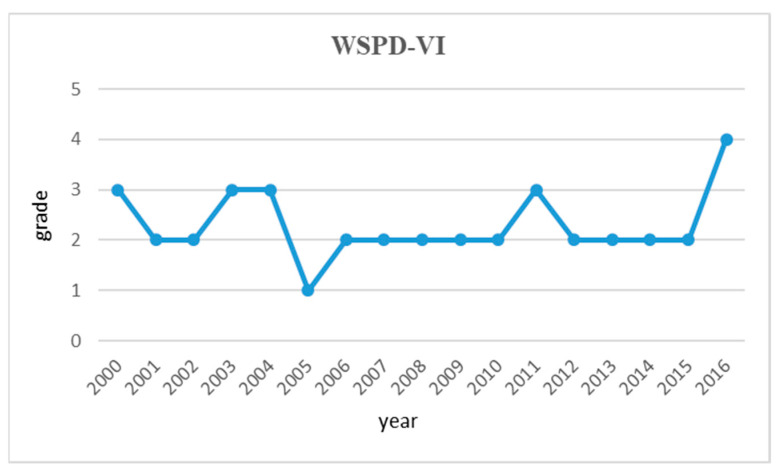
Trend graph of water resources vulnerability in the Huai River Basin during 2000~2016.

**Figure 6 entropy-21-00014-f006:**
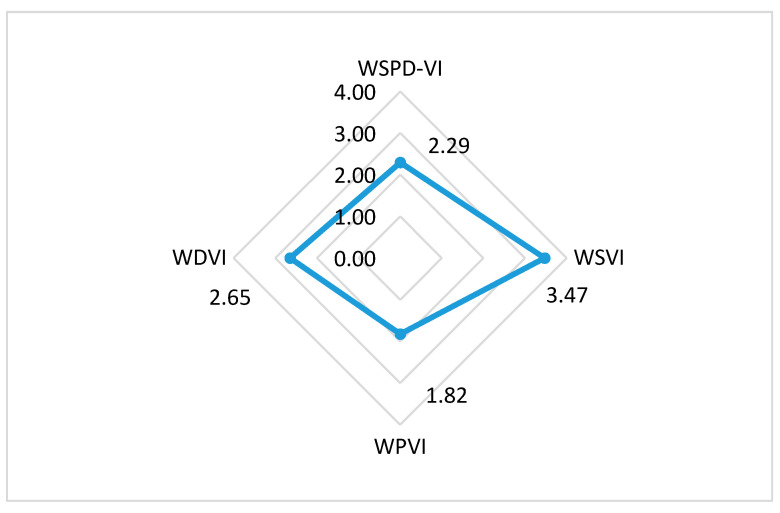
Radar chart of the water resources key vulnerability in the Huai River basin.

**Figure 7 entropy-21-00014-f007:**
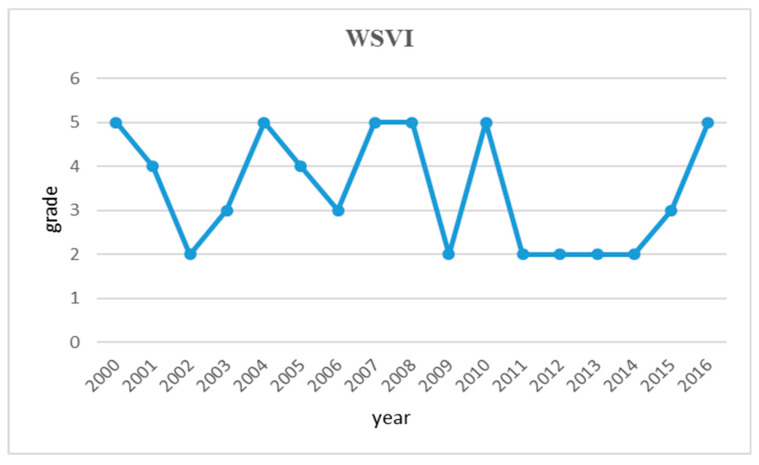
Trend graph of water shortage vulnerability in the Huai River Basin during 2000~2016.

**Figure 8 entropy-21-00014-f008:**
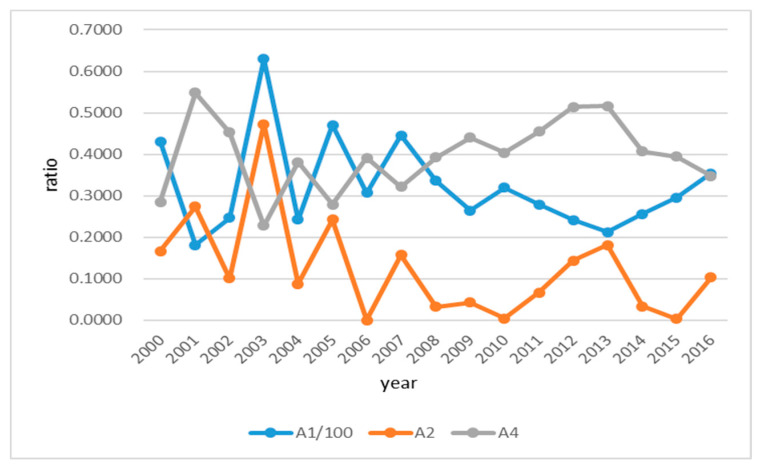
Trend graph of three water quantity evaluation indexes in the Huai River Basin during 2000~2016.

**Figure 9 entropy-21-00014-f009:**
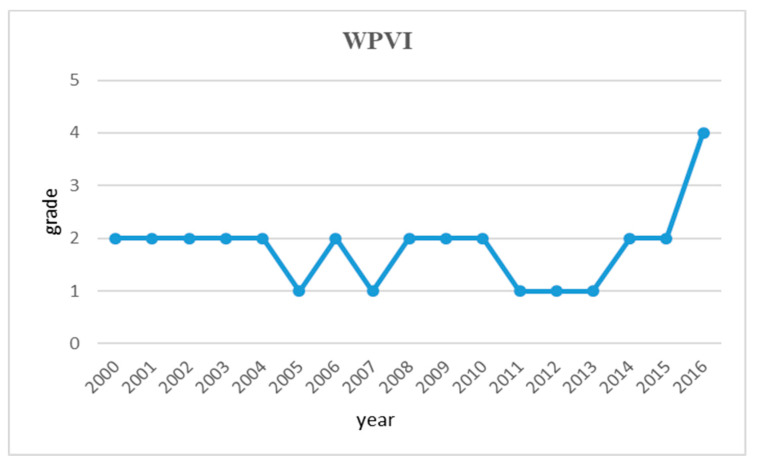
Trend graph of water pollution vulnerability in the Huai River Basin during 2000–2016.

**Figure 10 entropy-21-00014-f010:**
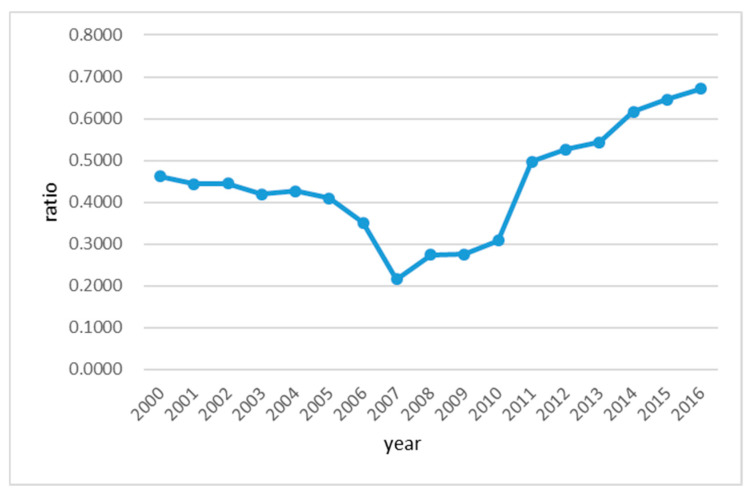
Trend graph of the water quality examination pass rate in water function area(*B*_1_) in the Huai River basin during 2000–2016.

**Figure 11 entropy-21-00014-f011:**
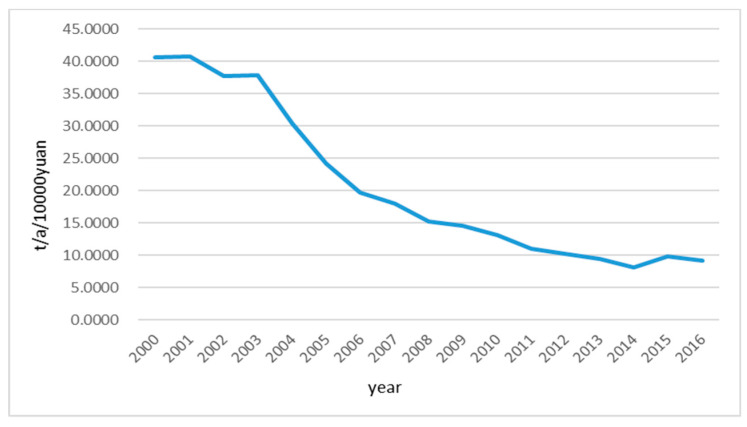
Trend graph of wastewater generation per 10,000-yuan GDP(*B*_6_) in the Huai River basin during 2000–2016.

**Figure 12 entropy-21-00014-f012:**
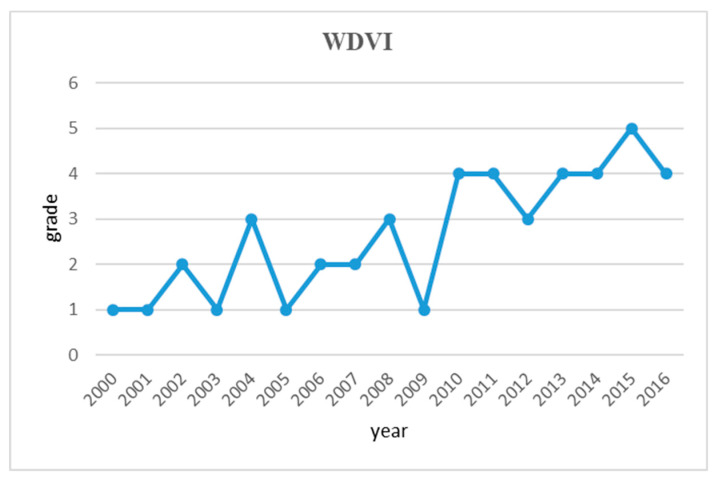
Trend graph of water-related natural disaster vulnerability in the Huai River basin during 2000–2016.

**Figure 13 entropy-21-00014-f013:**
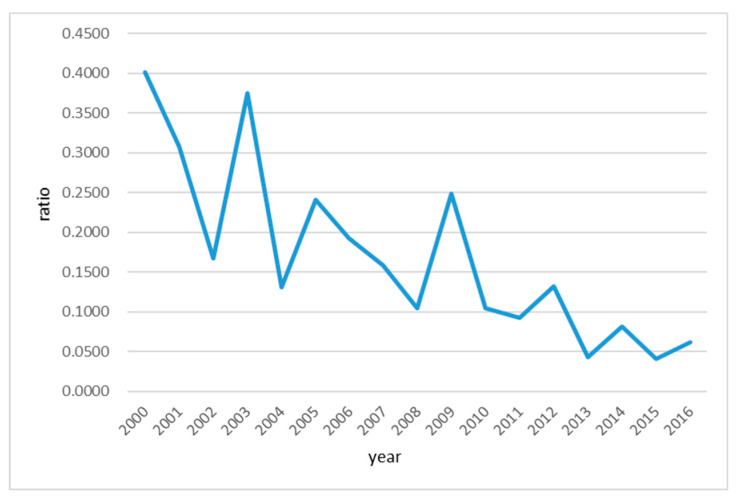
Trend graph of Proportion of area affected by flood and drought (*C*_2_) the ratio of flood and drought area to total area in the Huai River Basin during 2000~2016.

**Table 1 entropy-21-00014-t001:** Evaluation index system of river basin water resources vulnerability.

Evaluation Index System			Attribute Sign
Water shortage vulnerability (WSVI)	Natural vulnerability	Water yield per km^2^ *A*_1_	positive
		Absolute value of variation coefficient of annual precipitation *A*_2_	negative
	Man-made vulnerability	The proportion of surface water resources being utilized *A*_3_	negative
		The proportion of groundwater resources being utilized *A*_4_	negative
	Vulnerability of carrying capacity	Per capita water consumption *A*_5_	negative
		Water consumption for irrigation per mu *A*_6_	negative
Water pollution vulnerability (WPVI)	Natural vulnerability	Water quality examination pass rate in water function area *B*_1_	positive
		The decline rate of water quality examination pass rate *B*_2_	negative
	Man-made vulnerability	Total COD emission per 10,000 people *B*_3_	negative
		Total ammonia and nitrogen emission per 10,000 people *B*_4_	negative
	Vulnerability of carrying capacity	Ecosystem water consumption *B*_5_	positive
		Wastewater generation per 10,000-yuan GDP *B*_6_	negative
Water-related natural disaster vulnerability (WDVI)	Natural vulnerability	Water yield coefficient *C*_1_	positive
		Proportion of area affected by flood and drought *C*_2_	negative
	Man-made vulnerability	The proportion of soil erosion being controlled *C*_3_	positive
		The proportion of farmland area being the effectively irrigated *C*_4_	positive
	Vulnerability of carrying capacity	Proportion of population under levee protection *C*_5_	positive
		Regulation and storage capacity of water conservancy projects *C*_6_	positive

**Table 2 entropy-21-00014-t002:** The weights of the index system for evaluating the Huai River basin water resources vulnerability.

Evaluation Index System		Weight
WSVI	Water yield per km^2^ *A*_1_	0.0517
	Absolute value of variation coefficient of annual precipitation *A*_2_	0.1313
	The proportion of groundwater resources being utilized *A*_4_	0.1348
WPVI	Water quality examination pass rate in water function area *B*_1_	0.1454
	Total COD emission per 10,000 people *B*_3_	0.1371
	Wastewater generation per 10,000-yuan GDP *B*_6_	0.1422
WDVI	Water yield coefficient *C*_1_	0.0528
	Proportion of area affected by flood and drought *C*_2_	0.1380
	The proportion of soil erosion being controlled *C*_3_	0.0667

**Table 3 entropy-21-00014-t003:** Evaluation indicator standards for basin water resources vulnerability.

Indicator	Grading Standard of Basin Water Resource Vulnerability				
	Extreme Vulnerability (1st Level)	Severe Vulnerability (2nd Level)	Moderate Vulnerability (3rd Level)	Mild Vulnerability (4th Level)	Not Vulnerability (5th Level)
*A* _1_	0~17	17~25	25~32	32~40	40~60
*A* _2_	80%~100%	60%~80%	40%~60%	20%~40%	0~20%
*A* _4_	55%~100%	40%~55%	25%~40%	10%~25%	0~10%
*B* _1_	0~20%	20%~40%	40%~60%	60%~80%	80%~100%
*B* _3_	75~150	45~75	37.5~45	30~37.5	0~30
*B* _6_	8~30	6~8	4~6	2~4	0~2
*C* _1_	0~20%	20%~40%	40%~60%	60%~80%	80%~90%
*C* _2_	20%~40%	15%~20%	10%~15%	5%~10%	0~5%
*C* _3_	0~20%	20%~40%	40%~60%	60%~80%	80%~100%

**Table 4 entropy-21-00014-t004:** Degree of Huai River Basin water resources vulnerability between 2000 to 2016.

Year	WSPD-VI	WSVI	WPVI	WDVI
	Degree	Degree	Degree	Degree
2000	3	5	2	1
2001	2	4	2	1
2002	2	2	2	2
2003	3	3	2	1
2004	3	5	2	3
2005	1	4	1	1
2006	2	3	2	2
2007	2	5	1	2
2008	2	5	2	3
2009	2	2	2	1
2010	2	5	2	4
2011	3	2	1	4
2012	2	2	1	3
2013	2	2	1	4
2014	2	2	2	4
2015	2	3	2	5
2016	4	5	4	4
Mean value	2.29	3.47	1.82	2.65

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
