# Peer review of "Assessing Water Resources Vulnerability by Using a Rough Set Cloud Model: A Case Study of the Huai River Basin, China"

_entropy, 2018, doi:10.3390/e21010014_

Round 1

Reviewer 1 Report

The paper presents results of studies on the assessment of vulnerability of water resources to selected factors with the use of a cloud model. The Huai River Basin in Eastern China is examined as the case study. In my opinion, the submission is interesting and it may contribute to more precise determination of the volume and dynamics of water resources shortages, pollution and other in a river basin. However, there exist some shortcomings, which require improvements before the submission is accepted for publication. They are as follows:

1. There lacks detailed information of the input data used in the evaluation index system mentioned in Table 1.

2. The methodology of calculation of the elements applied in the model is complicated and thus not fully clear.

3. The content of Table 4 and Figures 5, 6 and 7 is the same, so, it can be reduced or completely deleted.

4. Figure 3 – description of the legend in the lower right corner is too small and thus unreadable.

5. Figure 4 – what is the meaning of numbers I, II, III, IV and V shown in the graphs?

6. Figure 8 – the vertical axis lacks description (units of the trend).

7. The English language used in the paper requires improvements, including grammar and orthography.

With the regard to the above-mentioned remarks it is recommended to accept the submission for publications after moderate revision.

Author Response

Response to Reviewer 1 Comments for

entropy-394153

“Assessing Water Resources Vulnerability by Using Rough Set Cloud Model: A Case Study of Huai River Basin, China”

General Comments:

The paper presents results of studies on the assessment of vulnerability of water resources to selected factors with the use of a cloud model. The Huai River Basin in Eastern China is examined as the case study. In my opinion, the submission is interesting, and it may contribute to more precise determination of the volume and dynamics of water resources shortages, pollution and other in a river basin. However, there exist some shortcomings, which require improvements before the submission is accepted for publication.

Response:

Thank you very much for your review and comments. We have followed all your comments as closely as possible to revise our manuscript. Your comments helped us improve the quality of the manuscript further. We greatly appreciate your help.

Point 1 There lacks detailed information of the input data used in the evaluation index system mentioned in Table 1.

Response 1:

Thank you for pointing out that. We have added some details about the data sources for input data to our evaluation index system. In detail, a new sentence had been added into the text between line 175 and 177: “the input data sources of the evaluation index system include the following: “The Water Resources Bulletin of the Basin”, “The China Environmental Statistics Yearbook”, and “The Chinese Statistical Yearbook”.”

In addition, detailed information of the input data can also be found between line 389-400.

Point 2 The methodology of calculation of the elements applied in the model is complicated and thus not fully clear.

Response 2:

Thank you for your comments. We have added some details about the methodology part to address your concern.

a)     First, we have added the detailed descriptions of the two methods that we have tried in pilot study but not actually adopted in final stage. Please see line 181 to 198 for details.

b)     Second, we have added one sentence between line 385 to 386 to highlight that the detailed work flow of our calculation is elaborated in Figure 2. This may help reader better understand our approach.

c)     Last but not least, we have carefully described how all the parameters that we used were obtained in section 2.4.2 and strengthened the details about our algorithm in section 3.2.1. In fact, we will also be glad to share our code for this study, which may offer a straight forward way to understand the calculation process in our approach.

Point 3 The content of Table 4 and Figures 5, 6 and 7 is the same, so, it can be reduced or completely deleted.

Response 3:

Thank you for the suggestions, but we are still willing to keep all these table and figures in our manuscript for the following reasons:

a)     We think that it is necessary to disclose all the details about the results of our water resource vulnerability calculation, which were presented in Table 4.

b)     To clearly demonstrate how water resource vulnerability have changed in Huai River basin during our research period, we have drawn a trend chart to specifically illustrate the trend for the comprehensive vulnerability. Instead of detailing the fact only by texts, we thought that Figure 5 is necessary to be kept in the context to help reader catch those facts quickly.

c)     Determination of key water vulnerabilities in Huai River basin is considered one of the crucial contributions of our study. Thus, we have calculated the mean values of each vulnerabilities to support such determination process and made a radar chart to visualize our findings in Figure 6. We thought that it is meaningful to keep this chart to give readers a clear idea on such issue through both descriptive and visualized means.

d)     To specifically demonstrate the trends of three sublevel water resources vulnerabilities during our study period, we have made trend charts for each of them, i.e. Figure 7, Figure 9, and Figure 12. We did think that it is necessary to keep those charts as data visualizations may help readers quickly catch the points that we wanted to illustrate. Therefore, we are still willing to keep Figure 7 in our manuscript, as well as Figure 9 and Figure 12.

We really appreciate that you can reconsider about our data visualization strategy and grant permissions for us to keep those figures in manuscript.

Point 4 Figure 3 – description of the legend in the lower right corner is too small and thus unreadable.

Response 4:

Thank you for pointing out that issue for us. We have enlarged the legend text in the lower right corner of Figure 3.

Point 5 Figure 4 – what is the meaning of numbers I, II, III, IV and V shown in the graphs?

Response 5:

Thank you for the question. In the figure 4, I, II, III, IV, V stands for “extreme vulnerability”, “severe vulnerability”, “moderate vulnerability”, “mild vulnerability”, and “not vulnerable”, respectively.

We have added some texts into the paragraph above the Table 3 and Figure 4 to illustrate the meaning of those Greek numbers. Please see line 489 to 491 for details.

Point 6 Figure 8 – the vertical axis lacks description (units of the trend).

Response 6:

Thank you for the comments. We have added a description of the vertical axis in Figure 8. The unit of such trend is a ratio, which ranges between 0 and 1. Corresponding details are also been added into the manuscript between line 573 to 575.

Point 7 The English language used in the paper requires improvements, including grammar and orthography.

Response 7: Thank you for the comments. We have made proof read and manuscript again and to correct many Grammar mistakes and syntax mistakes. In addition, we also conducted a new round of polishing on manuscript to increase the readability of our paper. We will continue to improve our works in terms of both English language usage and quality of contents.

Reviewer 2 Report

1.      The manuscript presents assessing the water resources vulnerability by using rough set cloud model with a case study of Huai River Basin, China, which is interesting. The subject addressed is within the scope of the journal.

2.      However, the manuscript, in its present form, contains several weaknesses. Appropriate revisions to the following points should be undertaken in order to justify recommendation for publication.

3.      Full names should be shown for all abbreviations in their first occurrence in texts. For example, WSPD-VI in p.1, RESC in p.2, etc.

4.      For readers to quickly catch your contribution, it would be better to highlight major difficulties and challenges, and your original achievements to overcome them, in a clearer way in abstract and introduction.

5.      It is shown in the reference list that the authors have a pertinent publication in this field. This raises some concerns regarding the potential overlap with their previous works. The authors should explicitly state the novel contribution of this work, the similarities and the differences of this work with their previous publications.

6.      It is mentioned in p.1 that a rough set cloud model is adopted to carry out the vulnerability evaluation. What are the other feasible alternatives? What are the advantages of adopting this particular model over others in this case? How will this affect the results? More details should be furnished.

7.      It is mentioned in p.3 that a water resource vulnerability evaluation approach combining rough set, information entropy, and cloud model is adopted in this study. What are the other feasible alternatives? What are the advantages of adopting this particular approach over others in this case? How will this affect the results? More details should be furnished.

8.      It is mentioned in p.3 that the improved blind deletion rough set method is adopted to carry out a dimension reduction on the full evaluation index system. What are the other feasible alternatives? What are the advantages of adopting this particular method over others in this case? How will this affect the results? More details should be furnished.

9.      It is mentioned in p.3 that the improved conditional information entropy rough set method is adopted to determine the weights of indexes. What are the other feasible alternatives? What are the advantages of adopting this particular method over others in this case? How will this affect the results? More details should be furnished.

10.  It is mentioned in p.4 that 18 evaluation variables are adopted in the evaluation index system. What are the other feasible alternatives? What are the advantages of adopting these particular variables over others in this case? How will this affect the results? More details should be furnished.

11.  It is mentioned in p.5 that “…If yes, we will continue to add the most important attributes of the set {??} to the set B by expert judgment…” How many experts are involved here? What are their specializations? What are their feedbacks? How to resolve conflicting feedbacks? How will these affect the results? More details and justifications should be furnished.

12.  It is mentioned in p.8 that MATLAB software is adopted to program the forward generator of cloud model. What are other feasible alternatives? What are the advantages of adopting this particular software over others in this case? How will this affect the results? The authors should provide more details on this.

13.  It is mentioned in p.8 that the Huai River basin is adopted as the case study. What are other feasible alternatives? What are the advantages of adopting this particular case study over others in this case? How will this affect the results? The authors should provide more details on this.

14.  It is mentioned in p.9 that historical records of 2000 to 2016 are taken. Why are more recent data not included in the study? Is there any difficulty in obtaining more recent data? Are there any changes to situation in recent years? What are its effects on the result?

15.  It is mentioned in p.11 that three methods are adopted to calculate the weight of evaluation indexes. What are the other feasible alternatives? What are the advantages of adopting these particular methods over others in this case? How will this affect the results? More details should be furnished.

16.  It is mentioned in p.14 that the mean annual values of the WSPD-VI, WSVI, WPVI, and WDVI are adopted to analyze key vulnerability of water resources in Huai River basin. What are the other feasible alternatives? What are the advantages of adopting this particular approach over others in this case? How will this affect the results? More details should be furnished.

17.  Some key parameters are not mentioned. The rationale on the choice of the particular set of parameters should be explained with more details. Have the authors experimented with other sets of values? What are the sensitivities of these parameters on the results?

18.  Some assumptions are stated in various sections. Justifications should be provided on these assumptions. Evaluation on how they will affect the results should be made.

19.  The discussion section in the present form is relatively weak and should be strengthened with more details and justifications.

20.  Moreover, the manuscript could be substantially improved by relying and citing more on recent literatures about contemporary real-life case studies of soft computing techniques in water resources management such as the followings:

l   Cheng, C.T., et al., “Flood control management system for reservoirs,” Environmental Modeling & Software 19 (12): 1141-1150 2004.

l   Fotovatikhah, F., et al., “Survey of Computational Intelligence as Basis to Big Flood Management: Challenges, research directions and Future Work,” Engineering Applications of Computational Fluid Mechanics 12 (1): 411-437 2018.

l   Taormina, R., et al., “Neural network river forecasting through baseflow separation and binary-coded swarm optimization”, Journal of Hydrology 529 (3): 1788-1797 2015.

l   Wu, C.L., et al., “Rainfall-Runoff Modeling Using Artificial Neural Network Coupled with Singular Spectrum Analysis”, Journal of Hydrology 399 (3-4): 394-409 2011.

l   Wang, W.C., et al., “Improved annual rainfall-runoff forecasting using PSO-SVM model based on EEMD,” Journal of Hydroinformatics 15 (4): 1377-1390 2013.

l   Chau, K.W., et al., “Use of Meta-Heuristic Techniques in Rainfall-Runoff Modelling” Water 9(3): article no. 186, 6p 2017.

21.  Some inconsistencies and minor errors that needed attention are:

l   Replace “…based the cloud model…” with “…based on the cloud model…” in line 135 of p.3

22.  In the conclusion section, the limitations of this study and suggested improvements of this work should be highlighted.

Author Response

Response to Reviewer 2 Comments for

entropy-394153

“Assessing Water Resources Vulnerability by Using Rough Set Cloud Model: A Case Study of Huai River Basin, China”

Point 1 and 2

1.      The manuscript presents assessing the water resources vulnerability by using rough set cloud model with a case study of Huai River Basin, China, which is interesting. The subject addressed is within the scope of the journal.

2.      However, the manuscript, in its present form, contains several weaknesses. Appropriate revisions to the following points should be undertaken in order to justify recommendation for publication.

Response 1 and 2:

Thank you for the comments, which are very helpful for improving the quality of our manuscript. We have addressed the problems and issue in our manuscript closely according to your comments and suggestions. We greatly appreciate your time and effort in reviewing this article and writing us the responses.

Point 3 Full names should be shown for all abbreviations in their first occurrence in texts. For example, WSPD-VI in p.1, RESC in p.2, etc.

Response 3:

Thank you for pointing out that. We have searched through the full text and replaced all the abbreviation in their first occurrence with the format of a full name following an abbreviation in parenthesis.

Point 4:  For readers to quickly catch your contribution, it would be better to highlight major difficulties and challenges, and your original achievements to overcome them, in a clearer way in abstract and introduction.

Response 4:Thank you for the suggestions, we have made the following revisions in our manuscript to address this problem:

            (1) We have rewritten the abstract to highlight the major contributions and significance of our study. Specifically, the second and third sentences of abstract have been changed to “However, as one of the major water systems in China, there is no existing evaluation index system can effectively assess water resource vulnerability for Huai River basin. To address this issue, we identified key vulnerability factors, constructed an evaluation index system, and applied such system to evaluate water resources vulnerability for Huai River basin empirically in this paper.”

            (2) We have specifically highlighted the major contributions of our works compared to existing literature in the introduction section. In fact, the major challenge to be addressed in this paper is to construct an evaluation index system that can assess the river basin water resource vulnerability effectively and comprehensively. Please see the line 121 to 128 in page 3 for details.

Point 5 : It is shown in the reference list that the authors have a pertinent publication in this field. This raises some concerns regarding the potential overlap with their previous works. The authors should explicitly state the novel contribution of this work, the similarities and the differences of this work with their previous publications.

Response 5:

Thank you for the suggestion. We have made some revisions in introduction section to specify how our works can contribute to literature in this filed. In detail, in line 118 to 121 in page 3, we have stated “However, the key vulnerability factor identification for the Huai River basin has not been done to our knowledge. Therefore, the key vulnerability factor identification of basin water resources has become an urgent problem to be solved in Huai River basin management.”

In addition, we also added some description to our manuscript to clarify the contribution of our study. Please see line 121 to 128 in page 3 for details.

Point 6 and 7:

6.      It is mentioned in p.1 that a rough set cloud model is adopted to carry out the vulnerability evaluation. What are the other feasible alternatives? What are the advantages of adopting this particular model over others in this case? How will this affect the results? More details should be furnished.

7.      It is mentioned in p.3 that a water resource vulnerability evaluation approach combining rough set, information entropy, and cloud model is adopted in this study. What are the other feasible alternatives? What are the advantages of adopting this particular approach over others in this case? How will this affect the results? More details should be furnished.

Response 6 and 7:

Thank you for raising these questions.

Based on our understanding, comment 6 and comment 7 are referring to the same concern on the general methodology that we adopted in this study. We have rewritten the first section of methodology part to address your concern on this issue. Please see line 147 to 156 in page 4 for details. The full text of such section is quoted below:

“In this paper, we firstly proposed an evaluation index system for water resources vulnerability evaluation. This comprehensive evaluation index system contains factors covering three different aspects: water shortage, water pollution, and water-related natural disasters. Then we carried out a dimension reduction on the full evaluation index system by using the improved blind deletion rough set method. Afterward, we used the improved conditional information entropy rough set method to determine the weights of indexes in such evaluation index system. Finally, based on the cloud model, we have evaluated the water resources vulnerability of Huai River basin, measured by calculated water resources vulnerability grading. Since we used rough set type approaches to conduct both weight determination and dimension reduction, and the final evaluation output were derived based on a cloud model, we name the general approach we use for this study as a rough set cloud model.”

Beyond the above explanation, we have also added some detailed information to explicitly explain the rationales of using evaluation index approach, improved blind deletion rough set method, conditional information entropy rough set method, and cloud model over their alternatives in section 2.1, 2.2, 2.3, and 2.4, respectively.

Point 8: It is mentioned in p.3 that the improved blind deletion rough set method is adopted to carry out a dimension reduction on the full evaluation index system. What are the other feasible alternatives? What are the advantages of adopting this particular method over others in this case? How will this affect the results? More details should be furnished.

Response 8:

Thank you for your comments. We have added a paragraph at the beginning of section 2.2 to specifically address your concern about the rationale behind choosing improved blind deletion rough set method over its alternatives. Please see line 181 to 198 for details.

Point 9 : It is mentioned in p.3 that the improved conditional information entropy rough set method is adopted to determine the weights of indexes. What are the other feasible alternatives? What are the advantages of adopting this particular method over others in this case? How will this affect the results? More details should be furnished.

Response 9:

Thank you for the comments. To clearly illustrate the rationale behind why we use improved conditional information entropy rough set method over its alternatives to determine the weights of indexes, we have added some details into the section 2.3.

Specifically, compared to common weight determination approaches such as AHP, using rough set-based approaches to determine the weights can effectively avoid the impacts of biased subjective judgements. In fact, we have performed a pilot study where we tried all three mainstream rough set approaches. We have added the technical details of all these three approaches in section 2.3, between lien 231 to line 252.

Additionally, we also added some explanation about the rationale that we choose the improved conditional information entropy rough set method over the other two that we also tried in pilot study in section 2.3. In fact, there are two reasons:

a)     First, since we are determining the weight for a reduced index system produced by the improved blind deletion rough set method, traditional rough set approach will automatically give consider some attributes as redundant and thereby assign zero weight to them, which include some of those attributes that were considered redundant by the model by manually retained by us. In another word, the traditional rough set weigh determination approach will delete some attributes that we intend to keep in the system. Therefore, we cannot use the traditional rough set approach.

b)     The conditional information entropy rough set approach will eventually produce the output with the weights of redundant attributes are greater than those considered nonredundant. This is certainty contradict to what we are intended to do.

Corresponding details are added to the manuscript between line 253 to 264.

Point 10 :  It is mentioned in p.4 that 18 evaluation variables are adopted in the evaluation index system. What are the other feasible alternatives? What are the advantages of adopting these particular variables over others in this case? How will this affect the results? More details should be furnished.

Response 10 :

At present, there are many index systems for water resources vulnerability assessment, which are designed from different perspectives, such as driving-pressure-state-response, risk-exposure-sensitivity-adaptability, etc.

In this study we constructed an evaluation index system for water resources vulnerability from three aspects: water shortage vulnerability (WSVI), water pollution vulnerability (WPVI), and water-related natural disaster vulnerability (WDVI), which are considered comprehensive and can include all the perspectives that are supposed to be incorporated by a typical study in this field.

In terms of the evaluation indicators used, we have incorporated some universal indicators that are commonly address in existing literature, such as water yield per km2, water quality examination pass rate in water function area, etc. The historical raw data for those universal indicators can be found from the published yearbooks. Meanwhile, we also designed some unique indicators in this study, such as A2, B2, B3, B4, B6, C2, C4, C5, and C6. These indicators are not directly cited from the statistical yearbook but are calculated combinations of several raw indicators. However, due to the space limit of this paper, the calculation process for those unique indicators are not specifically presented in the manuscript.

A total of 18 indicators are considered large number in this field, and a full evaluation system incorporating all of them is very comprehensive. There is no need to incorporate other alternative indicators into our system.

The above information is clearly elaborated in section 2.1.

Point 11: It is mentioned in p.5 that “…If yes, we will continue to add the most important attributes of the set {??} to the set B by expert judgment…” How many experts are involved here? What are their specializations? What are their feedbacks? How to resolve conflicting feedbacks? How will these affect the results? More details and justifications should be furnished.

Response 11:

Thank you for the comments. We have added appendix A into our manuscript to explain how we carried out the step 2 of improved blind deletion rough set method by adopting experts’ judgements. The detailed scoring table was also attached in that appendix.

In detail, we have included a total of 10 experts, who are Fuyong Zheng, Min Zhao, Sana, Taozhen Huang, Xiaoping Zhou, Yong Li, Qianqian Li, Xu Na, Guoping Tong, and Hongmei Chen. They are all working in the fields of water resources management, water conservation economy, and hydrology. However, we have no plan to further disclose their names to public due to privacy concern.

Point  12:  It is mentioned in p.8 that MATLAB software is adopted to program the forward generator of cloud model. What are other feasible alternatives? What are the advantages of adopting this particular software over others in this case? How will this affect the results? The authors should provide more details on this.

Response 12:

Thank you for the questions. Certainly, other software or programming languages such as R, Python, STATA can also be used to code our program. However, the results will not be affected. As long as the algorithm is the same and the programming language is a valid one, difference in programming language should not result in significant difference in final output. Since MATLAB has been widely regarded as a reliable way to conduct mathematical computing, we choose it as the software to adopt our algorithm.

Point 13: It is mentioned in p.8 that the Huai River basin is adopted as the case study. What are other feasible alternatives? What are the advantages of adopting this particular case study over others in this case? How will this affect the results? The authors should provide more details on this.

Response 13:

Thank you for the comments. First, our approach can be applied to any river basins in the world, will the evaluation results will certainly be different than this study. However, as what we have stated in various places in our manuscript, we are specifically focusing assessing the water resource vulnerability in Huai River basin in this study. This is because of following reasons:

a)     Huai River basin is one of the major river basins in China, which has greatly significance because of its importance in the country due to the population and economy in this area.

b)     No one had ever constructed an evaluation index approach that can evaluate the Huai River basin water resources vulnerability effectively. Thus, this study has filled this knowledge gap.

c)     It will be very meaningful to apply our approach to other river basin areas in China and other places in the world. But those application would require separate pilot studies and data collection process. Therefore, applying this approach to alternative research objectives would be entirely different empirical studies other than this one. Therefore, given current data availability and space limit of our paper, we focus on Huai River basin in this study.

The above reasons can be found in detail between line 375 to 383.

Point 14: It is mentioned in p.9 that historical records of 2000 to 2016 are taken. Why are more recent data not included in the study? Is there any difficulty in obtaining more recent data? Are there any changes to situation in recent years? What are its effects on the result?

Response 14:

Thank you for the question. The data that we used in this study is the most recent one that we can get. The reason for that is the yearbooks of water resource in China are published with a two-year lag. Therefore, as we wrote this paper in 2018, the most recent data we can got is 2016.

In addition, we choose the study period started from 2000 mainly because most of indicators that we used in this paper were started to be published in China since 2000. Before that there was no systematic survey of water resources in this area, thus there was no data available. Please see line 394 to 400 in page 11 for details.

The data availability is indeed a limitation of this study. However, we have mentioned this limitation and provided further outlook and solution for it in conclusion section, please see what we mentioned between line 778 to 785 for details.

Point 15: It is mentioned in p.11 that three methods are adopted to calculate the weight of evaluation indexes. What are the other feasible alternatives? What are the advantages of adopting these particular methods over others in this case? How will this affect the results? More details should be furnished.

Response 15:

Thank you for raising this question. In fact, we have tried all of these three methods to calculate the weight of evaluation indexes in a pilot study, and finally picked only the improved condition information entropy rough set method out of the three. To clarify this, we have changed the expression as “We use the improved condition information entropy rough set method to determine the weights of the evaluation index system.”

Please see our response to comment (9) for the detailed rationale behind how we pick this approach over its alternatives.

Point 16: It is mentioned in p.14 that the mean annual values of the WSPD-VI, WSVI, WPVI, and WDVI are adopted to analyze key vulnerability of water resources in Huai River basin. What are the other feasible alternatives? What are the advantages of adopting this particular approach over others in this case? How will this affect the results? More details should be furnished.

Response 16:

In addition to the mean annual values, we can also use the median as an alternative. But because our vulnerability index has only five levels, its median value will always be three, so that the median cannot reflect the differences between vulnerabilities.

On the other hand, using the mean annual values of WSPD-VI, WSVI, WPVI, and WDVI to analyze key vulnerability of water resources in Huai River basin can show a clear overall pattern of these indexes over our research period, and avoid the problem associated with using medians. Therefore, we thought that it is a simple but efficient way to keep using mean annual values to illustrate those details in our manuscript. Please see line 528 to 531 in page 15 for details

Point 17:  Some key parameters are not mentioned. The rationale on the choice of the particular set of parameters should be explained with more details. Have the authors experimented with other sets of values? What are the sensitivities of these parameters on the results?

Response 17:

Thank you for the comments, we have made some revision to explain how those parameters in the cloud model were determined in greater detail. The rationales behind the determination process are listed as follow:

1)   In formula we set the denominator to 2.355is based entirely on the experience of previous studies, and can be set to 6.

2)     In addition, for the value of He.  We set it to 0.01 in this study, but which is indeed subject to change. According to pervious literature, when 3*He is smaller than En, the characteristics of qualitative concepts can be better expressed. In this paper, we have tested various parameters an found that the parameter combination of Eni=2.3555 and He = K=0. 01 are best fitting the actual situation of the Huai River basin.

3)     In terms of the number of droplets needed, we have done multiple round of simulations and gradually increased the number of droplets from 3000 to 15000. The simulations show that a below-5000 number of droplets are not large enough to generate stable results, while the output becomes stable if the number of droplets is greater than 15000. Obviously, to balance the stable results and computational power needed, we decided to use 15000 droplets to construct the cloud model in this study.

Point 18:  Some assumptions are stated in various sections. Justifications should be provided on these assumptions. Evaluation on how they will affect the results should be made.

Response 18:

Particularly, we have two implied assumptions for our study:

a)     We assume that water resource vulnerability systems include only water quantity, water quality, and water-related natural disasters. No other parts of local ecosystems such as forest ecosystem, urban ecosystems, etc. have been included. This assumption is made to concentrate our focus of study on water system of Huai River basin and avoid intervenes from other perspectives.

b)     We assume that the rough set theory is also applicable to the water resources vulnerability assessment, that is, using the rough set method to reduce the dimension of the original index can remove the redundant information and will not cause the excessive loss of the data information according to the rough set method. As we have stated in the literature review, rough set have been widely adopted in water resource management fields. Therefore, this assumption should be valid here.

Point 19: The discussion section in the present form is relatively weak and should be strengthened with more details and justifications.

Response 19:

Thank you for the comments. We have added a Section 3.4 to specially discuss our main findings with strengthened details and justifications. Please see line 696 to 742 for details.

Point 20: Moreover, the manuscript could be substantially improved by relying and citing more on recent literatures about contemporary real-life case studies of soft computing techniques in water resources management such as the followings:

 Cheng, C.T., et al., “Flood control management system for reservoirs,” Environmental Modeling & Software 19 (12): 1141-1150 2004.

Fotovatikhah, F., et al., “Survey of Computational Intelligence as Basis to Big Flood Management: Challenges, research directions and Future Work,” Engineering Applications of Computational Fluid Mechanics 12 (1): 411-437 2018.

Taormina, R., et al., “Neural network river forecasting through base flow separation and binary-coded swarm optimization”, Journal of Hydrology 529 (3): 1788-1797 2015.

Wu, C.L., et al., “Rainfall-Runoff Modeling Using Artificial Neural Network Coupled with Singular Spectrum Analysis”, Journal of Hydrology 399 (3-4): 394-409 2011.

Wang, W.C., et al., “Improved annual rainfall-runoff forecasting using PSO-SVM model based on EEMD,” Journal of Hydroinformatics 15 (4): 1377-1390 2013.

Chau, K.W., et al., “Use of Meta-Heuristic Techniques in Rainfall-Runoff Modelling” Water 9(3): article no. 186, 6p 2017.

Response 20:

Thank you for the suggestions. We have cited those articles you mentioned and added some contents into our literature review section. Please see line 94 to 102 for details.

Point 21: Some inconsistencies and minor errors that needed attention are:

Replace “…based the cloud model…” with “…based on the cloud model…” in line 135 of p.3

Response 21:

Thank you for pointing out that, we have carefully proof-read the manuscript again to fix those errors and inconsistencies.

Point 22: In the conclusion section, the limitations of this study and suggested improvements of this work should be highlighted.

Response 22:

Thank you for the comments. We have clearly stated the limitation of this study and possible further improvement in the last paragraph of conclusion section. The primary limitation of this study is availability of data, which has to be addressed by collecting more data observations in the future. Thus, we proposed that similar study might be repeated as more data appears. In addition, if resources permit, we will also be willing to conduct such study again in the future. Please see line 778 to 785 for details.

Reviewer 3 Report

This research involves unclear scientific approaches and approximation to solve the addressed problems. Therefore, I cannot accept this manuscripts for the publication.

Author Response

Response to Reviewer 3 Comments for

entropy-394153

“Assessing Water Resources Vulnerability by Using Rough Set Cloud Model: A Case Study of Huai River Basin, China”

Point :

This research involves unclear scientific approaches and approximation to solve the addressed problems. Therefore, I cannot accept this manuscript for the publication.

Response:

We are so sorry to see that you made such as decision to decline our submission. However, we still hope you can reconsider about your decision as the methodology of this study have been widely adopted in various fields. We believe that our study would be a meaningful supplement to the water resource management literature for the following reasons:

a)    This study filled the knowledge gap that to evaluate the Huai River basin water resource vulnerability through a quantitative methodology. As one of the major river basins in China with huge population and vast area, Huai River basin has been facing the water resources problems for a long time. As a foundation of constructing an effective local water management system, it is necessary to carry out a water resources vulnerability evaluation for this region. However, no exiting literature has addressed this issue in the past.

b)    We have polished our texts again to clearly show the structure of our study in section 2. Specifically, we firstly proposed an evaluation index system for water resources vulnerability evaluation. This index system contains factors covering three different aspects: water shortage, water pollution, and water-related natural disasters. Then we carried out a dimension reduction on the full evaluation index system by using the improved blind deletion rough set method. Afterward, we used the improved conditional information entropy rough set method to determine the weights of indexes in such evaluation index system. Finally, based on the cloud model, we have evaluated the water resources vulnerability of Huai River basin, measured by calculated water resources vulnerability grading. Since we used rough set type approaches to conduct both weight determination and dimension reduction, and the final evaluation output were derived based on a cloud model, we name the general approach we use for this study as a rough set cloud model. Figure 2 also presented a visualization of our work flow for this study, which is pretty clear and easy to be understand.

c)    The methodology we use for this study has been widely adopted in various fields. Such as following references:

[1] Han L, Li C, Liu H. Feature Extraction Method of Rolling Bearing Fault Signal Based on EEMD and Cloud Model Characteristic Entropy[J]. Entropy, 2015, 17(10):6683-6697.

[2] Haobo Zhang, Yunna Wu ,A Method for Multi-Criteria Group Decision Making with 2-Tuple Linguistic Information Based on Cloud Model[J].Information, 2017, 8, 54:

[3]Binbin Shi,Wei Wei ,A Novel Energy Efficient Topology Control Scheme Based on a Coverage-Preserving and Sleep Scheduling Model for Sensor Networks[J].Sensors, 2016, 16, 1702;

[4] Risk Evaluation of a UHV Power Transmission Construction Project Based on a Cloud Model and FCE Method, for Sustainability[J]. 2015.Sustainability, Vol. 7, Pages 2885-2914:

[5] Ke-Qin Wang , Hu-Chen Liu .Green Supplier Evaluation and Selection Using Cloud Model Theory and the QUALIFLEX Method.[J]Sustainability.2017, 9688

[6] Qiuyan Liu,Mingwu Wang, Land Eco-Security Assessment Based on the Multi-Dimensional Connection Cloud Model [J]Sustainability 2018, 10, 2096

[7] Shengmei Yang, Xianquan Han,Cloud-Model-Based Method for Risk Assessment of Mountain Torrent Disasters.[J].Water 2018, 10, 830

[8] Maryam Zavareh,Viviana Maggioni,Application of Rough Set Theory to Water Quality Analysis: A Case Study.[J]Water.a 2018, 3, 50

[9]Jingqian Wang , Xiaohong Zhang, Four Operators of Rough Sets Generalized to Matroids and a Matroidal Method for Attribute Reduction.[J].Symmetry 2018, 10, 418

Those successful applications of similar or same methodology to our study are solid proof that both rough set and cloud model are valid and reliable scientific approaches, which can be employed to address our interested topic for this study.      

Round 2

Reviewer 2 Report

The revised paper has addressed all my previous comments, and I suggest to ACCEPT the paper as it is now.